# Strong anion exchange-mediated phosphoproteomics reveals extensive human non-canonical phosphorylation

Gemma Hardman[1,†], Simon Perkins[2,†], Philip J Brownridge[1], Christopher J Clarke[1], Dominic P Byrne[3], Amy E Campbell[1], Anton Kalyuzhnyy[2], Ashleigh Myall[2], Patrick A Eyers[3] (iD), Andrew R Jones[2] & Claire E Eyers[1,*] (iD)

## Abstract

Phosphorylation is a key regulator of protein function under (patho)physiological conditions, and defining site-specific phosphorylation is essential to understand basic and disease biology. In vertebrates, the investigative focus has primarily been on serine, threonine and tyrosine phosphorylation, but mounting evidence suggests that phosphorylation of other "non-canonical" amino acids also regulates critical aspects of cell biology. However, standard methods of phosphoprotein characterisation are largely unsuitable for the analysis of non-canonical phosphorylation due to their relative instability under acidic conditions and/or elevated temperature. Consequently, the complete landscape of phosphorylation remains unexplored. Here, we report an *unbiased* *p*hospho-peptide enrichment strategy based on strong *a*nion exchange (SAX) chromatography (UPAX), which permits identification of histidine (His), arginine (Arg), lysine (Lys), aspartate (Asp), glutamate (Glu) and cysteine (Cys) phosphorylation sites on human proteins by mass spectrometry-based phosphoproteomics. Remarkably, under basal conditions, and having accounted for false site localisation probabilities, the number of unique non-canonical phosphosites is approximately one-third of the number of observed canonical phosphosites. Our resource reveals the previously unappreciated diversity of protein phosphorylation in human cells, and opens up avenues for high-throughput exploration of non-canonical phosphorylation in all organisms.

**Keywords** mass spectrometry; non-canonical phosphorylation; phosphohistidine; phosphoproteomics; strong anion exchange chromatography

**Subject Categories** Methods & Resources; Post-translational Modifications & Proteolysis

**The EMBO Journal (2019) 38: e100847**

## Introduction

It is very well established that post-translational modification (PTM) of proteins by phosphorylation rapidly and reversibly regulates cellular signalling functions, impinging on catalytic activity, protein:protein/DNA/RNA interactions, protein localisation and stability (Cohen, 2002; Hunter, 2012). Moreover, protein phosphorylation is an essential PTM that has been identified across the kingdoms of life and whose dysregulation in human cells is closely associated with diseases such as cancer and diabetes. Until recently, phosphorylation of the hydroxyl-containing amino acids serine (Ser), threonine (Thr) and tyrosine (Tyr) was thought to be the primary mode of phosphorylation-mediated signalling in non-plant eukaryotes. However, a growing body of evidence (Besant & Attwood, 2012; Fan *et al*, 2012; Fraczyk *et al*, 2015; Fuhs *et al*, 2015; Shen *et al*, 2015; Wieland & Attwood, 2015; Panda *et al*, 2016; Srivastava *et al*, 2016; Xu *et al*, 2016a,b; Fuhs & Hunter, 2017) indicates that phosphorylation of other amino acids, termed here "non-canonical" phosphorylation, including His, Asp, Glu, Lys, Arg and Cys [in addition to pyrophosphorylation of Ser and Thr to form ppSer and ppThr (Chanduri *et al*, 2016; Harmel & Fiedler, 2018), and Lys polyphosphorylation (Bentley-DeSousa & Downey, 2019)], may also regulate protein signalling functions. Of note, the generation of both site-specific and generic antibodies against phosphohistidine (pHis) (Kee *et al*, 2013, 2015; Fuhs *et al*, 2015; Lilley *et al*, 2015) has recently allowed mammalian cell-type independent roles for this PTM to be elucidated (Fuhs *et al*, 2015) in processes as diverse as ion channel regulation (Srivastava *et al*, 2016) and T-cell signalling (Panda *et al*, 2016), and during cell proliferation, differentiation and migration. In contrast, and despite occasional reports in the literature (Besant *et al*, 2009; Attwood *et al*, 2011), the prevalence (or otherwise) of other non-canonical protein phosphorylation events has not been reliably confirmed in human cells, and so specific information about the extent and details of the sites of modification is lacking.

1 Centre for Proteome Research, Department of Biochemistry, Institute of Integrative Biology, University of Liverpool, Liverpool, UK
2 Department of Comparative and Functional Genomics, Institute of Integrative Biology, University of Liverpool, Liverpool, UK
3 Department of Biochemistry, Institute of Integrative Biology, University of Liverpool, Liverpool, UK
*Corresponding author. Tel: +44 0151 795 4424; E-mail: ceyers@liverpool.ac.uk
†These authors contributed equally to this work

Unlike relatively stable phosphate esters (pSer, pThr, pTyr), the phosphoramidate bond in pHis, pLys and pArg, and the phosphorothioate bond of pCys are highly susceptible to hydrolysis at low pH and/or at elevated temperature (Hultquist *et al*, 1966; Hultquist, 1968; Stock *et al*, 1990; Attwood *et al*, 2007). Consequently, traditional (bio)chemical techniques used to characterise sites of canonical Ser/Thr/Tyr phosphorylation, including acid-based phosphopeptide enrichment prior to high-throughput phosphoproteomics analysis, are largely unsuitable for the analysis of "atypical" phosphorylated amino acids due to extensive phosphate hydrolysis. Loss of the phosphate group before the site of peptide modification can be pinpointed, means that acid-labile phosphopeptides are dramatically under-represented in the literature (Gonzalez-Sanchez *et al*, 2013). As a result, cellular understanding of the roles of non-canonical protein phosphorylation in vertebrates lags far behind that of canonical pSer, pThr and pTyr. Focussed analytical tools are therefore essential to help redress this imbalance (Lasker *et al*, 1999; Attwood, 2013; Gonzalez-Sanchez *et al*, 2013; Fuhs & Hunter, 2017) and to permit the functional roles of individual modified residues to be defined in proteomes.

To help tackle this significant knowledge gap, we have developed UPAX, an *u*nbiased *p*hosphopeptide enrichment strategy based on strong *a*nion e*x*change (SAX) chromatography, which permits enrichment of both canonical and non-canonical (pX) phosphorylated peptides at near-neutral pH (pH 6.8). Using this UPAX workflow, we report ~1,300 novel sites of non-canonical phosphorylation on His, Asp, Glu, Lys, Arg and Cys residues in human proteins using tandem mass spectrometry (MS/MS). This new resource reveals that the number of sites of pHis, pLys and pArg is each of a similar order to the numbers observed for pTyr under basal conditions, while the total number of non-canonical phosphorylation sites is roughly a third of that observed for all non-canonical phosphorylation, even having accounted for likely false site localisation. This wealth of novel phosphorylation site information allows us to define amino acid motifs flanking non-canonical phosphosites, and to exploit enrichment of gene ontology (GO) terms and UniProt keywords to evaluate the potential function of proteins enriched in specific non-canonical phosphorylated residues.

# Results

## Standard phosphopeptide enrichment strategies are largely unsuitable for acid-labile phosphopeptides

Although pH-dependent phosphate hydrolysis is established for free phosphorylated amino acids including pHis (Hultquist, 1968), and for pHis-containing peptides at low pH (~2.5) (Potel *et al*, 2018b) to our knowledge, the relative stability of acid-labile protein-derived pHis-containing peptides over a range of pH values has not been reported. To quantify pH stability and generate standards for optimisation of enrichment strategies for acid-labile phosphorylated peptides, we produced pHis-containing tryptic peptides by proteolysis of chemically phosphorylated myoglobin (Figs EV1 and EV2 and Appendix Table S1). The two isomers of pHis, 1-pHis and 3-pHis, exhibit different stabilities due to the difference in $\Delta G°$ of hydrolysis of the phosphoramidate bond; we ascertained by dot

blot with pHis isomer-specific antibodies (Fuhs *et al*, 2015) that the chemically generated pMyo preferentially contained 3-pHis (Fig EV1A). At pH 1, we observed extensive hydrolysis of pHis from myoglobin-derived phosphopeptides ($t_{1/2}$ values of ~15 min), although the modification was more tolerant to mildly acidic conditions than has been previously assumed, with a $t_{1/2}$ at pH 4 of > 2 h, comparable to that observed at near-neutral pH values (Appendix Fig S1). Rapid dephosphorylation was also observed for these pHis peptides at elevated temperatures, with the $t_{1/2}$ at 95°C being similar to that at pH 1 (~15 min), while at 80°C, the $t_{1/2}$ increased to 45–60 min.

The relative stability of these pHis peptides at pH 4 prompted us to compare standard phosphoproteomics enrichment strategies, including (i) TiO$_2$ enrichment under a variety of binding, wash and elution condition; (ii) calcium phosphate precipitation (minimising time spent at low pH); and (iii) hydroxyapatite (HAP) chromatography, quantifying enrichment and recovery for the individual pHis peptides. For each strategy, recovery of pHis peptides was quantified and compared with the enrichment of canonical monoester α-/β-casein-derived pSer/pThr peptides (see Appendix Supplementary Methods). Irrespective of the method employed, enrichment of pHis-containing peptides was highly inefficient in our hands, even under conditions where pSer/pThr peptide enrichment was optimal (Appendix Tables S2–S4), primarily due to pHis hydrolysis during enrichment and/or sample clean-up (Appendix Fig S2), or, in the case of HAP chromatography, inefficient peptide recovery (Appendix Table S4). These purification procedures are therefore deemed unsuitable for enrichment of acid-labile phosphopeptides prior to LC-MS/MS analysis, highlighting the need for a different approach for efficient isolation of non-canonical phosphopeptides for MS/MS analysis.

## Strong anion exchange enables separation of acid-labile pHis-containing peptides

Ion exchange chromatography, in which peptides are separated based on differences in charge, is used extensively in LC-MS/MS-based proteomics (Wolters *et al*, 2001). Strong anion exchange (SAX) is eminently suited to phosphopeptide separation, given the presence of a negatively charged phosphate group, including a second formal negative charge that is acquired above pH ~6 (Han *et al*, 2008; Alpert *et al*, 2015). However, phosphopeptide elution from SAX columns is typically performed by decreasing the pH, which will compromise the stability of acid-labile phosphopeptides. To assess the effect of ionic strength (as opposed to pH)-mediated elution for phospho(His)peptide analysis, we employed a modified triethylammonium phosphate gradient (Alpert *et al*, 2015) for chromatographic separation at pH 6.0, 6.8 and 8.0. As shown in Fig 1A–C, the majority of phosphopeptide ion signals from tryptic digests of myoglobin (pHis) and casein (pSer/pThr) eluted in later SAX fractions, irrespective of pH, while non-phosphorylated peptides dominated the earlier fractions. Comparative fractionation of phosphopeptide mixtures by high-pH reverse-phase chromatography confirmed that differential chromatographic elution of phosphopeptides from non-phosphopeptides is not simply an effect of peptide separation, but arises due to charge-based differences in affinity for the SAX solid phase. Extracted ion chromatograms (XICs) of myoglobin pHis peptides and their

non-phosphorylated counterparts revealed that some pHis peptides underwent phosphate hydrolysis over the chromatographic time-course when SAX was performed at pH 6.0 (Fig EV3). Optimal separation of pHis peptides from their non-phosphorylated counterparts was preferentially observed at pH 6.8 (Figs 1B and EV3); peptide signal in the latter SAX fractions derived almost entirely from phosphopeptides in these relatively simple mixtures. Crucially, spiking phosphorylated myoglobin into a highly complex human cell lysate did not affect separation of pHis myoglobin peptides by SAX under these conditions (Fig 1D). Moreover, LC-MS/MS analysis confirmed that recovery of pHis myoglobin peptides was similar to that of non-phosphorylated and pSer/pThr peptides present in the "spiked" mixture. Together, these data validate UPAX at pH 6.8 for the separation of acid-labile pHis peptides from complex mixtures for subsequent analysis by LC-MS/MS.

## UPAX-based phosphoproteomics of HeLa cell extracts

With the intention of increasing "basal" levels of non-canonical phosphorylation prior to UPAX and phosphoproteome analysis (Fig 2A), we employed siRNA to abrogate expression of PHPT1, one of three reported mammalian phosphohistidine phosphatases (Ek *et al*, 2002; Klumpp *et al*, 2002; Hindupur *et al*, 2018), which can also act as a pLys phosphatase *in vitro* (Ek *et al*, 2015). Immunoblotting of cell extracts confirmed efficient knockdown of PHPT1 in HeLa cells (Fig 2B), from which UPAX was subsequently performed. Sixteen SAX fractions were collected and analysed by LC-MS/MS using HCD nlEThcD, where precursor ions exhibiting neutral loss of 98 amu following HCD were subjected to EThcD fragmentation (Frese *et al*, 2013; Ferries *et al*, 2017). A representative UV trace for the SAX separation (Fig 2C) and associated base peak

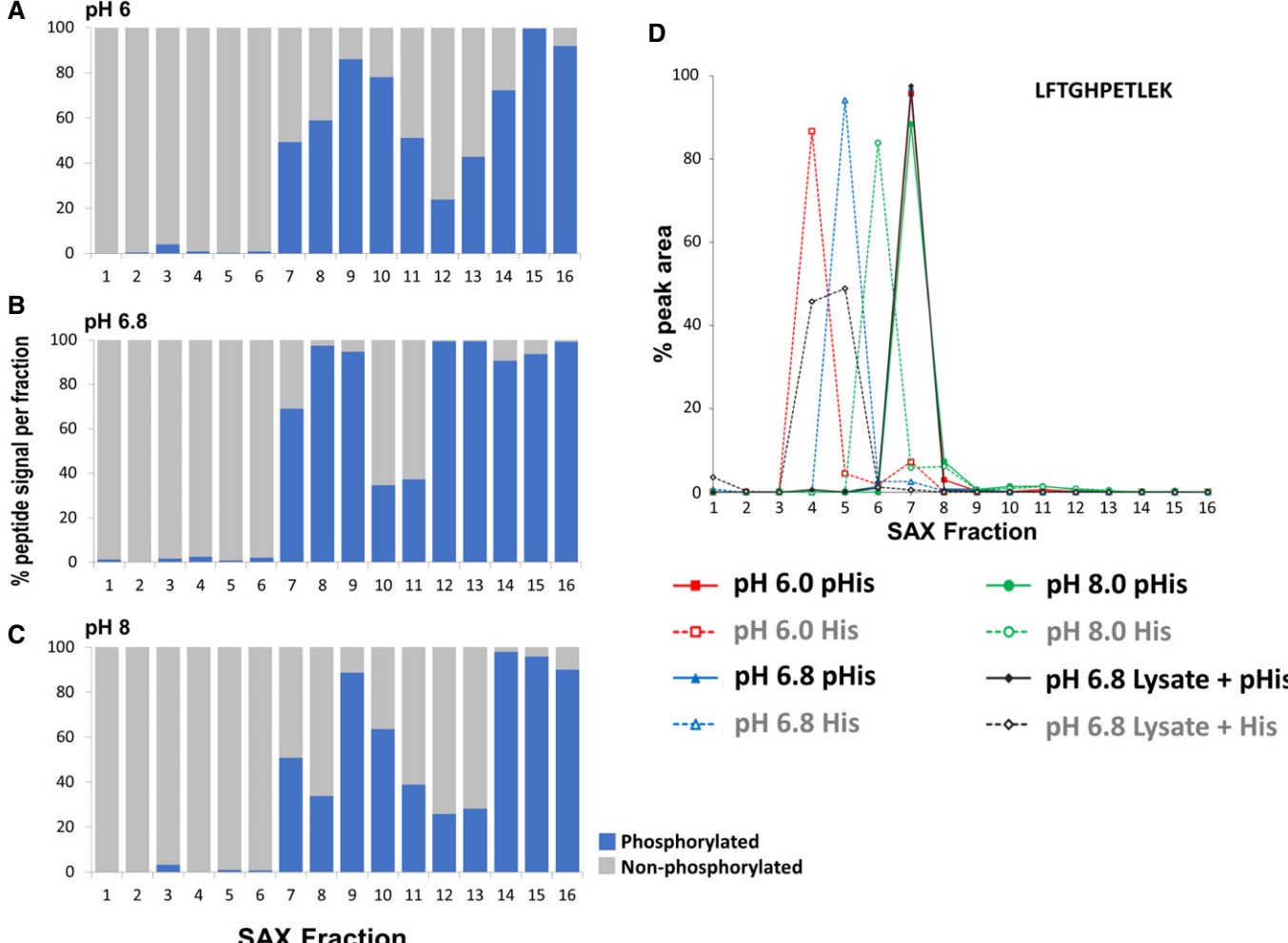

**Figure 1. Optimal separation of pHis myoglobin peptides from their non-phosphorylated counterparts by strong anion exchange (SAX) is achieved at pH 6.8.**

A–C Percentage of the total signal intensity attributed to phosphopeptides (blue bars, 11 unique peptide ions) and non-phosphorylated peptides (grey bars, 25 unique peptide ions) following SAX fractionation at (A) pH 6.0, (B) pH 6.8 and (C) pH 8.0, with later fractions consisting of up to 100% phosphopeptide ion signal.

D Representative pHis-containing myoglobin peptide (LFTGHPETLEK, solid line) and its non-phosphorylated counterpart (LFTGHPETLEK, dashed line), quantified across all 16 SAX fractions; SAX was performed at pH 6.0 (red), pH 6.8 (blue) or pH 8.0 (green). Percentage of the total peak area of each individual peptide across the gradient is plotted for each fraction. In order to assess pHis stability and effects of SAX separation in a complex mixture (black), SAX was also repeated at pH 6.8 with phosphorylated myoglobin spiked into a human cell lysate prior to digestion. Data for the five pHis-containing myoglobin peptides are shown in Fig EV3.

chromatograms for three of the 16 pooled SAX fractions are shown (Fig 2D–F). All MS/MS data analysis was performed using Proteome Discoverer, invoking MASCOT for peptide identification and *ptm*RS to evaluate phosphosite localisation confidence.

## Phosphorylation of His, Asp, Glu, Lys, Arg and Cys amino acids is readily detected in human cell extracts

Identification of non-canonical phosphorylation sites on human proteins required us to search our UPAX MS/MS data considering variable phosphorylation of all possible canonical and non-canonical amino acids. However, this type of analysis is very challenging in terms of statistical power; for any given peptide, the number of candidates produced in the overall peptide database is governed by $2^n$, where $n$ = number of modification sites being considered. If no variable modifications are included in a search, each peptide sequence gives rise to a single candidate ($2^0 = 1$). An "all-pX" search, which considers phosphorylation on Ser, Thr, Tyr, His, Asp, Glu, Lys, Arg, and/or Cys, involves all tryptic peptides having $n \gg 0$. For example, a peptide with 9 potential sites of phosphorylation (canonical and non-canonical: STYDECHKR) will contribute 512 candidates to the database. Consideration of a large

number of possible isobaric PTM sites thus raises several problems. First, it gives low statistical power when attempting to identify the peptide sequence from a given spectrum. Second, even if a peptide sequence can be identified with confidence, it is especially difficult to achieve high confidence in localisation of a modification site. Third, previously validated approaches for calibration of PTM site localisation confidence (defining false localisation rates, FLRs), which are based on comparison of scores against standard peptide libraries with known PTM sites, are not directly applicable (see below).

To streamline the process of simultaneously searching the full dataset with multiple isobaric variable modifications, a limited human database was employed (7,254 entries *c.f.* 20,187 sequences in the human UniProt database), comprising only those proteins identified from a first pass search in which only variable Ser/Thr/Tyr phosphorylation was considered. In total, >70,500 phosphopeptides (5% PSM FDR) were identified across all SAX fractions when considering 9 potential phosphorylatable residues, irrespective of site localisation confidence. Consistently, and as observed with phosphorylated myoglobin and casein peptide standards, the vast majority of these phosphopeptides derived from the latter SAX fractions (Appendix Fig S3). Applying the "class I" *ptm*RS score cut-off

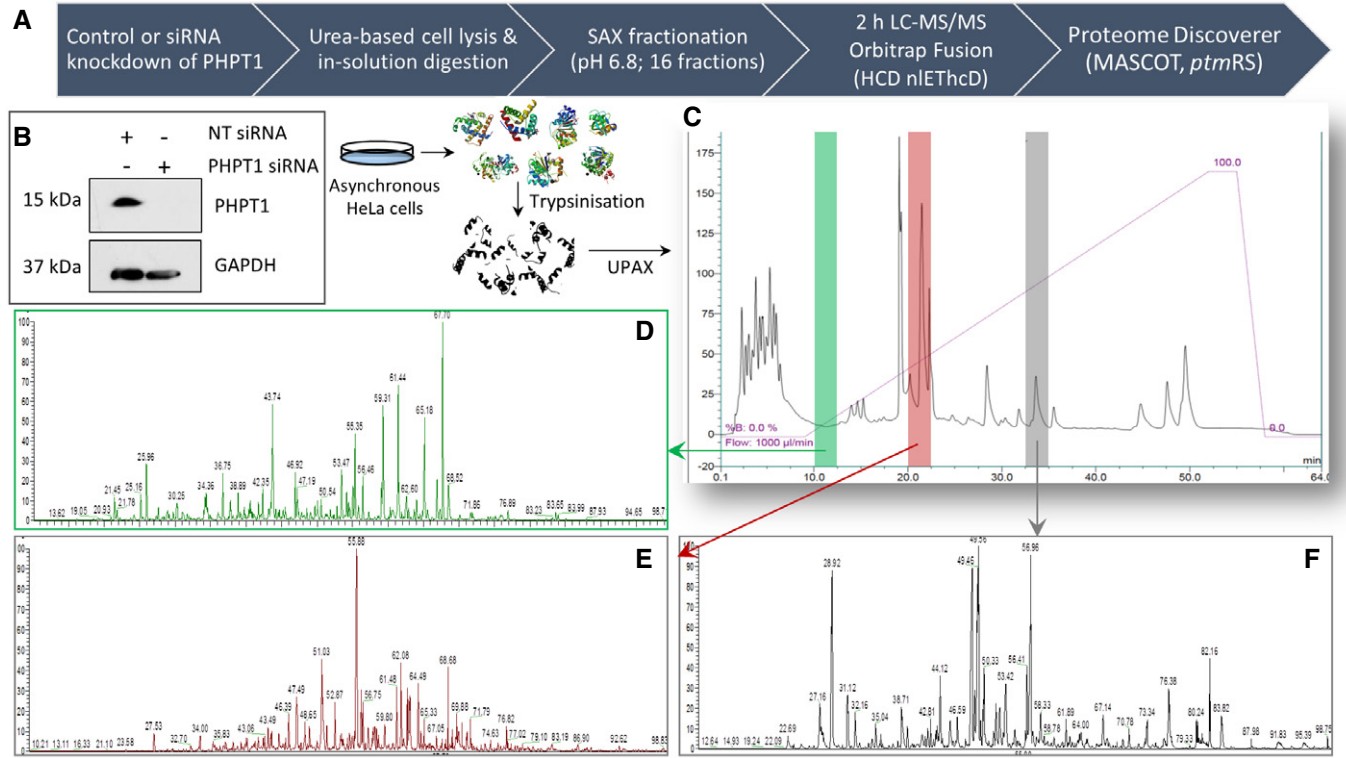

**Figure 2. UPAX workflow for unbiased canonical and non-canonical phosphoproteomics.**

A    Schematic representation of the UPAX strategy for phosphopeptide enrichment and MS/MS analyses.

B    siRNA-mediated knockdown of PHPT1 in HeLa cells (24 h) with reference to GAPDH loading control. NT—non-targeting siRNA control.

C–F  (C) Representative SAX profile of trypsin-digested HeLa lysate (Abs$_{280\ nm}$). Base peak chromatograms are shown for select SAX fractions following high-resolution LC-MS/MS using an Orbitrap Fusion mass spectrometer: (D) fraction 3 (green); (E) fraction 6 (red); and (F) fraction 10 (grey). Peptides were fragmented by HCD, with neutral loss of 98 amu from the precursor ions triggering EThcD (Ferries *et al*, 2017). Tandem mass spectra were separated according to fragmentation strategy in Proteome Discoverer prior to searching with Mascot. The ptmRS node was used for phosphosite localisation. Analysis was performed on three independent biological replicates for each condition (NT or PHPT1 siRNA).

of 0.75, which is typically invoked in high-throughput phosphoproteomics datasets, yielded ~12,500 pSer-containing peptide spectrum matches (PSMs), 1,750 pThr PSMs and 399 pTyr PSMs (Appendix Table S5). Remarkably, using this UPAX workflow, 320 PSMs were also defined as containing pHis, only slightly lower than the total number defined for pTyr, a very well established, but relatively rare canonical phosphorylation event. When considering all other non-canonical phosphorylated residues, the vast majority of PSMs were defined as containing pAsp (1,611) and pGlu (1,722); 844, 567 and 126 PSMs were further defined as containing pLys, pArg and pCys, respectively (Appendix Table S5). These PSMs correlated with ~3,400 unique novel non-canonical phosphosites from these human cell extracts: 225 for pHis sites; 980 for pAsp; 1,068 for pGlu; 626 for pLys; 419 for pArg; and 103 for pCys, compared with 3,636, 895 and 236 phosphosites for pSer, pThr and pTyr, respectively (Fig 3A and B and Datasets EV1 and EV2). However, as discussed in the following section, there is likely to be considerable false discovery amongst these identifications. Contradictory to expectations, knockdown of the pHis phosphatase PHPT1 did not result in a gross increase in the numbers of pHis peptides identified. In contrast, there was a marginal decrease in the number of phosphopeptides identified irrespective of the residue type, suggesting that the functions of PHPT1 as a pHis (or potentially pLys) phosphatase are likely to be redundant.

For comparison, we applied the same LC-MS/MS acquisition strategy and data interrogation pipeline to analyse a $TiO_2$-enriched phosphopeptide set prepared using a more "typical" phosphoproteome workflow (Ferries *et al*, 2017) from HeLa cells (Fig 3C and Appendix Table S6). While ~41% of the phosphorylation sites identified at *ptm*RS 0.75 were assigned to non-canonical residues in the UPAX dataset, this decreased to 4.4% in the phosphopeptides enriched using $TiO_2$, demonstrating the necessity to maintain near-neutral pH during sample preparation for the identification of non-canonical phosphosites.

## A method for estimation of false localisation rate (FLR)

To address the possibility that consideration of multiple phosphorylated residues compromises the statistical confidence for specific phosphosite identification, we used the same search parameters to evaluate the prevalence of the theoretical residue pAla, by substituting variable modification of pCys for pAla. Ala cannot be phosphorylated and so functions here as an internal control for search parameter evaluation and estimation of reside-specific false localisation rates (FLRs). Apparent pAla assignments (at different *ptm*RS thresholds) were used to estimate the rate at which the informatics workflow randomly assigns a phosphosite to a given amino acid (pX), having normalised for the relative frequency of each amino acid (X) being considered; i.e., a rare amino acid would have fewer chances of being randomly (and wrongly) assigned as a phosphosite than a common amino acid (see Appendix Supplementary Methods, Appendix Table S8). Such an alternate method of estimating FLR is particularly important for correct interpretation of the phosphoproteomics data presented in this resource, as all current phosphosite localisation tools, including *ptm*RS, at best have only been benchmarked against synthetic phosphopeptide libraries that are (i) singly phosphorylated, and/or (ii) only contain canonical pSer, pThr or pTyr. Consequently, none are optimally designed to evaluate such a

vast increase in number of isobaric PTMs, meaning that FLR could be significantly underestimated.

Initially, we verified that searching for pAla produced similar counts of PSMs at 5% FDR and checked the distribution of *ptm*RS scores for pAla "identification" versus other pX residues, demonstrating a similar distribution of scores (Appendix Fig S4). *ptm*RS typically gives a trimodal distribution with peaks close to 0.5, 0.85–0.9 and 1. Based on our simulations, we determined that the mode centred at 0.5 is caused by PSMs having identical ions between the 1st and 2nd ranked identified phosphosites. The mode centred around *ptm*RS 0.85 arises due to rank 1 PSMs having a single additional fragment ion match than the rank 2 site, while the mode at, or close to, *ptm*RS = 1 arises due to PSMs of rank 1 having more than one fragment ion match than the rank 2 site candidate. The width of a given distribution is governed by the length of a peptide. We thus conclude that pAla follows the same *ptm*RS distribution as other amino acids, allowing us to calibrate FLR estimates, i.e. estimate the rate at which a given amino acid is randomly designated as being phosphorylated (see Appendix Supplementary Methods, Appendix Tables S5, S7 and S8).

Perhaps not unexpectedly, given the high relative abundance of Ala in proteins, interrogating phosphopeptide PSMs for theoretical pAla (0.75 *ptm*RS) revealed 404 "identifications", almost double the number of identified pTyr sites (Appendix Table S5). At a *ptm*RS value of ≥ 0.75, the pAla decoy-computed FLR for pSer was 21%, while the FLR for pTyr was 41%. The overall mean FLR for all canonical and non-canonical phosphosites was 47%. Reducing the acceptable *ptm*RS value to 0.90 decreased the absolute number of "identified" pAla sites to 162, and improved false localisation rates, so that the pAla-estimated pSer FLR was 13%, while FLRs for pThr and pTyr were 26 and 28%, respectively (5% FDR, *ptm*RS ≥ 0.90; Fig 4A and Appendix Table S5). Inevitably, false localisation reduced further with an increase in acceptable *ptm*RS value to 0.99 (6% for pSer, 14% pThr and 17% for pTyr), although there was a ~2.1–2.5-fold reduction in the total number of assigned canonical phosphosites (Fig 3 and Appendix Table S5).

Increasing the PSM FDR to 1% predictably reduced the number of unique phosphopeptides identified (Fig EV4 and Appendix Table S7), but did not significantly improve the overall FLR: for pSer, the pAla-computed FLR decreased from 13 to 10% with a change in FDR from 5 to 1%, while the FLR for pTyr and pHis increased to 31 and 42%, from 28 and 36%, respectively, at a 1% FDR (Appendix Fig S5, Appendix Tables S5 and S7). Consequently, we conclude that the classically employed "class I" *ptm*RS value of 0.75 was unacceptable for the analysis of such complex datasets, with a *ptm*RS score of 0.90 (5% PSM FDR) yielding an optimal return, balancing false negatives with false-positive site identifications. Using this approach, we are now able to provide an estimate of the number of "true-positive" identifications for each pX residue (Fig 4B).

To validate our approach for residue-specific FLR estimation by pAla searching, we evaluated overlap between pTyr sites catalogued in PhosphoSitePlus (Hornbeck *et al*, 2015) and our identified pTyr sites, filtered to contain only those pTyr sites observed on peptides that did not also contain a site of non-canonical phosphorylation (given the likelihood that these sites would not have been identified in previous studies where non-canonical phosphorylation was not considered). This analysis revealed an overlap of 52% (61 out of

**A**

| ptmRS ≥ 0.75 | pSer | pThr | pTyr | pHis | pAsp | pGlu | pLys | pArg | pCys |
|---|---|---|---|---|---|---|---|---|---|
| Total pX peptides | 12543 | 1759 | 399 | 329 | 1611 | 1722 | 351 (844) | 403 (567) | 126 |
| Unique pX sites | 3636 | 895 | 236 | 225 | 980 | 1068 | 268 (626) | 278 (419) | 103 |
| ptmRS ≥ 0.90 | pSer | pThr | pTyr | pHis | pAsp | pGlu | pLys | pArg | pCys |
| Total pX peptides | 7409 | 1028 | 215 | 163 | 589 | 611 | 161 (513) | 191 (282) | 70 |
| Unique pX sites | 2317 | 489 | 138 | 129 | 410 | 427 | 140 (406) | 139 (220) | 55 |
| ptmRS ≥ 0.99 | pSer | pThr | pTyr | pHis | pAsp | pGlu | pLys | pArg | pCys |
| Total pX peptides | 3165 | 456 | 83 | 45 | 145 | 153 | 46 (232) | 66 (116) | 25 |
| Unique pX sites | 1105 | 210 | 51 | 37 | 99 | 112 | 45 (191) | 47 (91) | 20 |

**B**

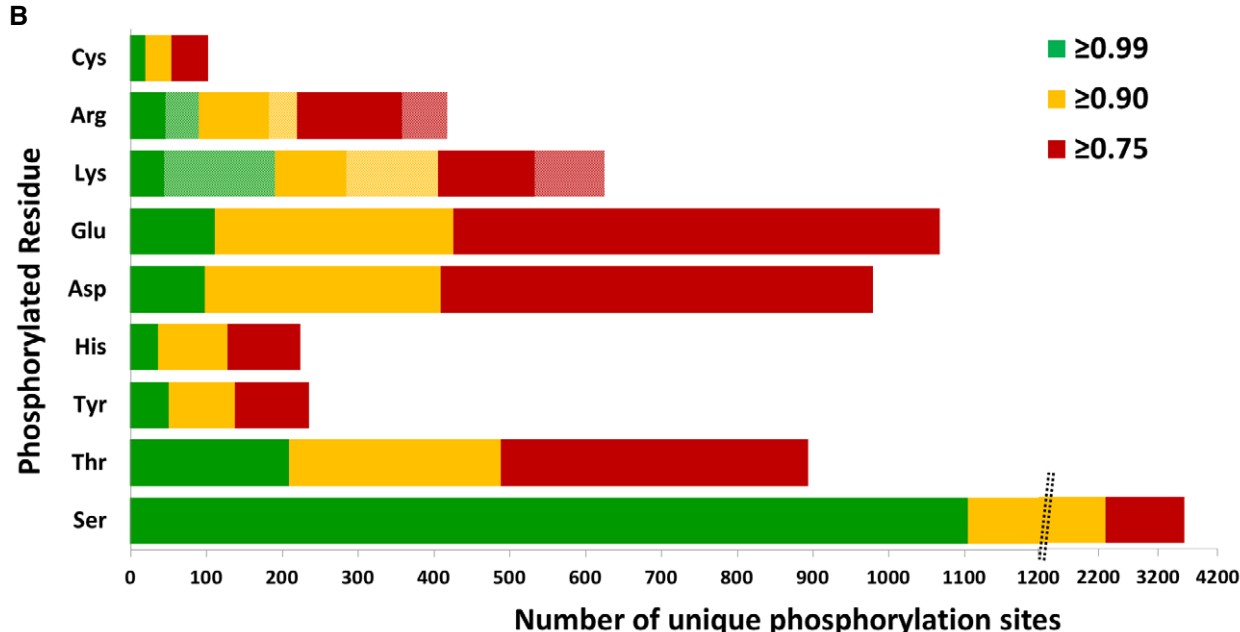

**C**

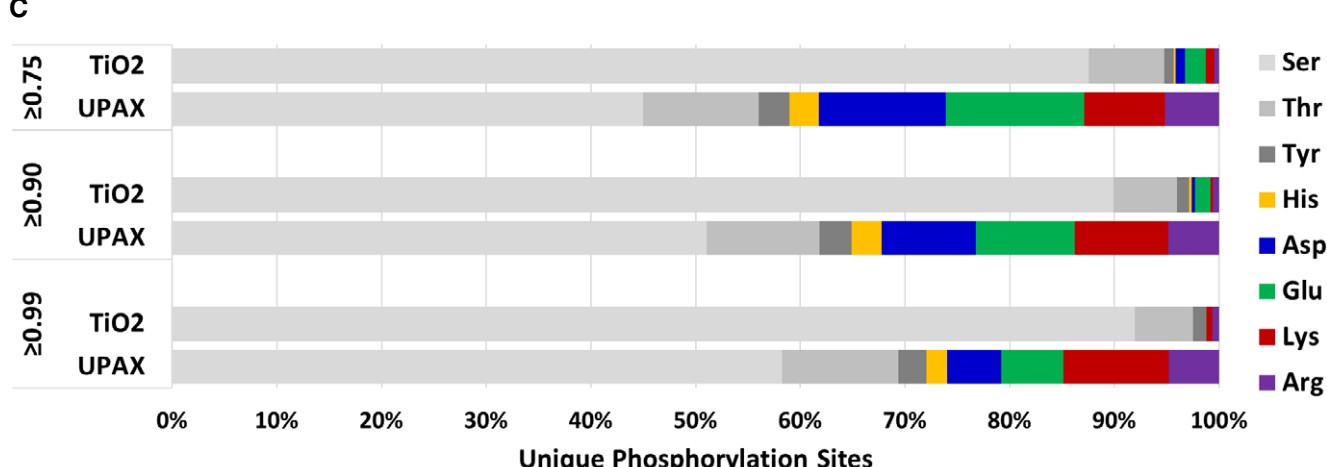

Figure 3.

**Figure 3.  UPAX permits identification of extensive phosphorylation of canonical and non-canonical phosphosites.**

A  Total number of phosphopeptides identified (5% FDR) and unique sites for each phosphorylated residue according to site localisation confidence (ptmRS score). For pLys and pArg, the number in parentheses is the total number of identified sites/peptides including those localised to the peptide C-terminus; outside of parentheses are the non-C-terminally mapped pLys or pArg sites.

B  Number of unique phosphorylation sites defined at different site localisation confidence values: ptmRS ≥ 0.99 (green); ptmRS ≥ 0.90 (yellow); and ptmRS ≥ 0.75 (red). Light green, yellow or red indicates those sites of pLys or pArg mapped to the extreme peptide C-terminal residue at each ptmRS score cut-off.

C  Percentage of canonical pSer, pThr and pTyr sites (shades of grey) compared with non-canonical phosphosites (pHis—yellow; pAsp—blue; pGlu—green; pLys—red; pArg—purple) identified using either the described UPAX strategy or a standard $TiO_2$-based phosphopeptide enrichment protocol, as a function of ptmRS score.

118 unique singly phosphorylated pTyr-containing peptides), giving a maximum FLR estimate of ~48% based on previous discovery studies. Although higher than the 28% FLR estimated from the pAla decoy search (which may also be, in part, due to incomplete coverage of the pTyr constituent of the phosphoproteome in the database), this pAla decoy analysis provides an improved measure of confidence of FLR estimation than implementation of *ptm*RS alone, one of the most commonly used phosphosite localisation tools in high-throughput phosphoproteomics studies.

**Residue-specific ions do not improve confidence in identification of non-canonical phosphopeptides**

In order to evaluate potential pX-specific marker ions that might increase confidence in identification of phosphopeptides containing a given phosphorylated amino acid, we interrogated HCD spectra for pX-specific phosphoimmonium ions. Immonium ions present in high-energy CID spectra can be considered good markers for the presence of a particular (modified) amino acid within a peptide sequence. As predicted, we observed a pTyr immonium ion at $m/z$ 216.04, although this was present in fewer spectra than expected, being observed in 14.5% of confidently localised pTyr-containing peptides (*ptm*RS ≥ 0.99), decreasing to 10.7% at *ptm*RS ≥ 0.90. These findings are broadly in agreement with a previous report stating that this pTyr immonium ion was only observed in ~20% of HCD spectra from pTyr peptides when present in a mixture with other types of phosphopeptides, although the rationale for the decrease in pTyr immonium ions in extremely complex mixtures has yet to be explained (Potel *et al*, 2018a). Moreover, although the presence of an ion at $m/z$ 216.04 is generally deemed indicative of the presence of pTyr, 5.0% (± 2.7% dependent on the nature of the phosphorylated residue) of HCD spectra from all other pX-containing phosphopeptides also contained an ion at this $m/z$ value. Even at a *ptm*RS value of 0.99, 4.0% of all spectra defined as containing phosphosites other than pTyr contained a product ion at $m/z$ 216.04. Furthermore, and in contrast to previous findings reported by Lemeer and colleagues (Potel *et al*, 2018b), the pHis immonium ion at $m/z$ 190.04 was not observed (> 5% relative signal intensity) in any pHis-containing peptides, irrespective of the *ptm*RS score (Appendix Table S9). Interrogation of HCD spectra containing other non-canonical residues also failed to identify significant proportions of phosphoresidue-specific immonium ions. Together, these data suggest that the presence, or lack thereof, of phosphoimmonium ions is not indicative of phosphorylation of any particular residue in this type of data, including pTyr.

It has been reported that pHis peptides exhibit a unique gas-phase fragmentation "fingerprint" when subjected to resonance

collision-induced dissociation (CID), arising as a result of neutral loss of 80 ($PO_3$), 98 ($PO_3 + H_2O$) and 116 ($PO_3 + 2H_2O$) atomic mass units (amu) from the precursor ion via mechanisms distinct from that of pSer neutral loss (Oslund *et al*, 2014). Oslund *et al* proposed that these ions arise due to loss of (i) the phosphate group (Δ80 amu) or (ii) phosphoric acid via nucleophilic attack of the acyl phosphate formed with the α-carboxyl group at the peptide C-terminus (Δ98 amu). Additional loss of $H_2O$ is also possible, most likely from the side chains of Asp or Glu, but in some instances from backbone oxygens. Irrespective of this, it was hypothesised that this precursor "triplet" neutral loss pattern could be exploited to improve identification of pHis-containing peptides from LC-MS/MS data. To investigate the utility of "triplet" neutral loss in these HCD-driven analyses as a signature for pHis peptide identification in high-throughput phosphoproteomics, we evaluated HCD-induced precursor neutral loss from pHis peptide ions following UPAX-LC-MS/MS of a complex digested human cell lysate (Fig 5). To characterise other pX residue-specific neutral loss characteristics, we also examined the prevalence of these precursor neutral loss ions from phosphopeptides containing each of the different canonical or non-canonical phosphorylated residues according to site localisation confidence (*ptm*RS score).

As shown in Fig 5A, neutral loss incidence was similar overall for phosphopeptides where the site of modification was confidently localised to Asp, Glu, Lys or Arg (*ptm*RS ≥ 0.90). Approximately 25% of all these phosphopeptides exhibited no precursor ion neutral loss, marginally higher than the 19 and 20% observed respectively for pHis- and pTyr-containing peptides. Unlike the relatively abundant and "specific" triplet loss pattern previously reported from pHis-containing peptides ions subject to resonance CID (~40% triplet loss for pHis precursor peptide ions compared with only 5% for pSer/pThr peptides), only 17% of all singly phosphorylated pHis-containing peptides (0.90 *ptm*RS) exhibited neutral loss of all three ions upon HCD, compared with ~26% of pSer/pThr peptides (Fig 5A). Of note, 16% of those phosphopeptide HCD spectra identified as containing pAla also exhibited this triplet neutral loss pattern, slightly higher than the 11% of pLys peptides with losses of 80, 98 and 116 amu. Thus, the triplet "signature" appears to be much less specific for pHis peptides subjected to HCD-mediated fragmentation than has been reported previously in a small dataset following CID.

We also evaluated the propensity for precursor ion neutral loss arising from EThcD. Although there was a higher likelihood of precursor ion neutral loss from all types of phosphopeptides (except those containing pHis), the relative abundance of those phosphopeptide precursor ions observed to exhibit neutral loss of all three groups (80, 98 and 116 amu) was much lower (Fig 5B). However, care should be taken when drawing conclusions from these

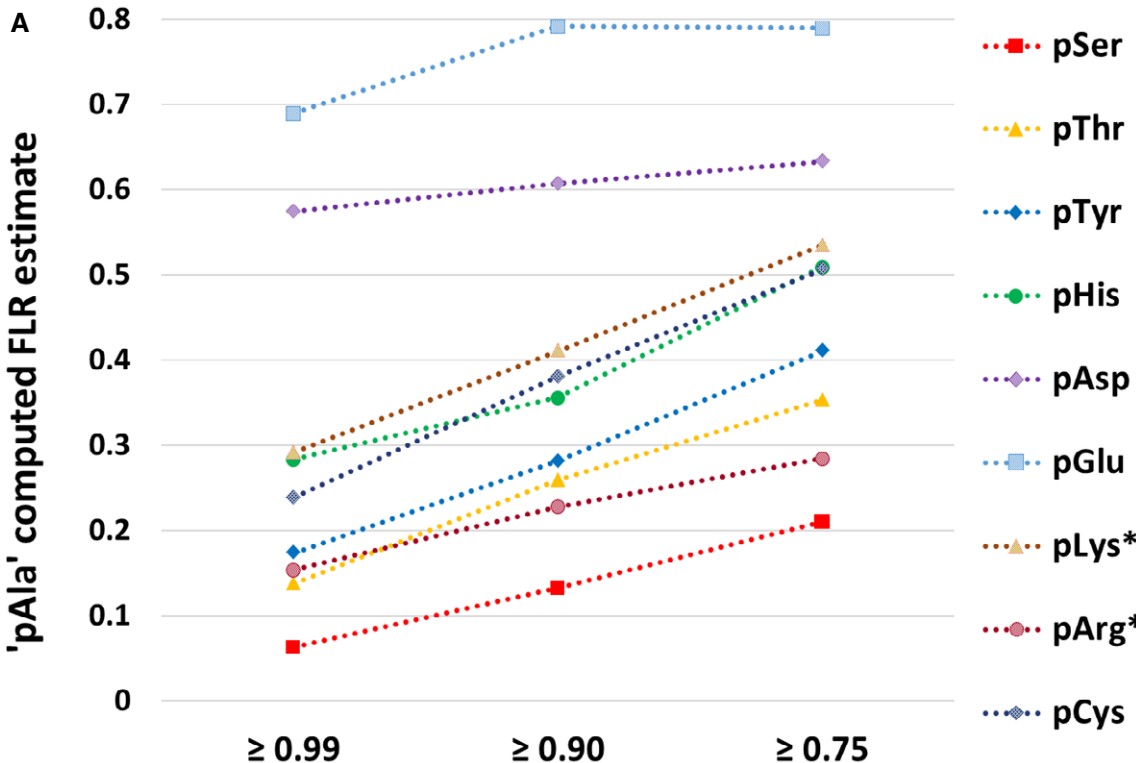

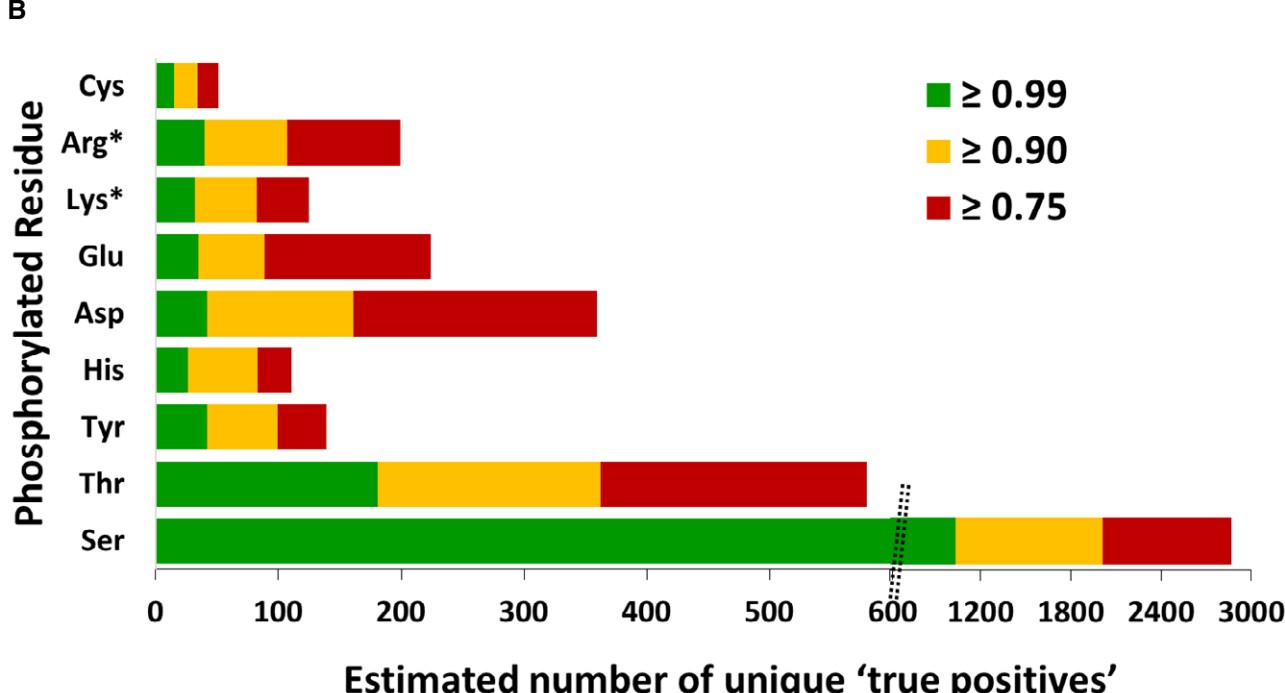

**Figure 4. A decoy pAla search can be used to estimate residue-specific false localisation rate and thus the number of identified true-positive phosphosites.**

A   pAla-computed FLR estimate for each type of canonical and non-canonical phosphorylated residue at ptmRS values ≥ 0.99, ≥ 0.90 or ≥ 0.75.

B   Total number of phosphorylation sites estimated to be "true positives" based on the pAla-determined residue-specific FLR, for each phosphorylated residue according to ptmRS score. pLys* and pArg* represent non-C-terminal pLys or pArg residues, respectively (see Appendix Supplementary Methods).

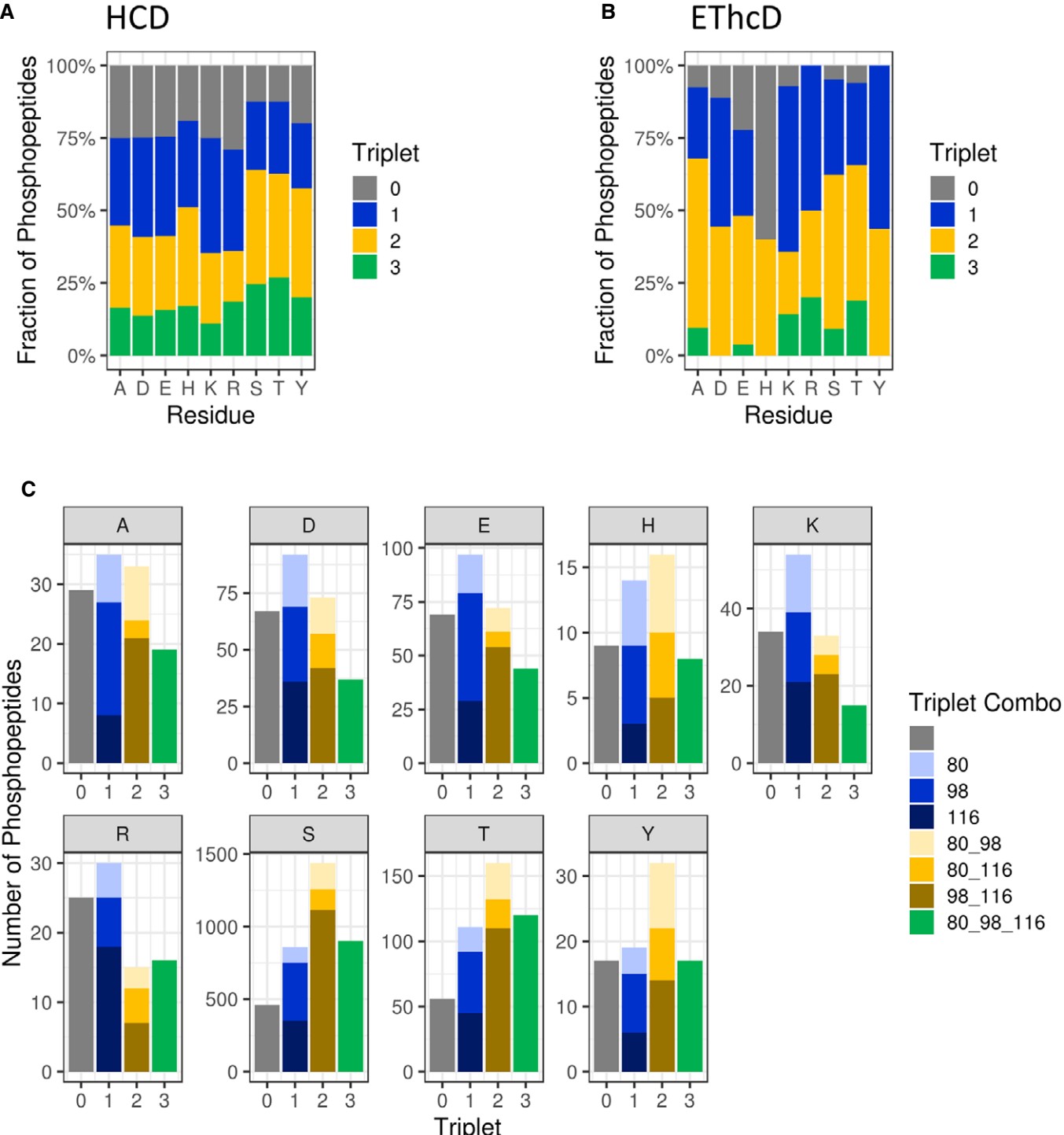

**Figure 5. Phosphopeptide neutral loss pattern upon HCD is not diagnostic for phosphorylation of any given residue.**

A   Comparison of triplet score (≥ 5% S/N) following HCD of unique phosphopeptides (5% FDR) identified from UPAX separated tryptic digests of HeLa cell-derived phosphopeptides. Site of phosphorylation (ptmRS score ≥ 0.90) is indicated. Percentage of phosphopeptides exhibiting either no neutral loss (triplet = 0) or neutral loss of any 1 (triplet = 1), 2 (triplet = 2) or 3 ions from the precursor (Δ80, Δ98 and/or Δ116 amu).

B   Distribution of triplet score following HCD fragmentation as a function of site localisation confidence (ptmRS score ≥ 0.75, ≥ 0.90, ≥ 0.95 and ≥ 0.99) for unique singly phosphorylated peptide as a function of the residue phosphorylated.

C   Numbers of phosphopeptides (ptmRS ≥ 0.90) exhibiting different combinations of the three neutral loss species (Δ80, Δ98 and Δ116 amu from precursor) for each of the phosphorylated residues.

observations considering the comparatively small numbers of (particularly non-pSer) singly phosphorylated peptides identified by EThcD (only 216 of which were not pSer) compared with HCD in this study (of which 1,352 were not assigned as pSer).

A detailed examination of the types of neutral loss ions generated is presented in Fig 5C. The relative proportion of pThr, pTyr and pHis peptides that lost $HPO_3$ (42, 46 and 51%, respectively) was proportionally higher than was identified for the other types of phosphopeptides (~28–34%), including pAla-containing decoys (Fig 5C). However, with the exception of the predicted preference for loss of phosphoric acid (with or without additional $H_2O$) from pSer and pThr peptides (68–71% for $H_3PO_4$ and 66–69% for $H_3PO_4 + H_2O$), no general trends were observed. However, we did note an increased prevalence for loss of two water molecules in addition to $HPO_3$, i.e. 116 amu (54%) for pArg peptides in comparison with the other non-canonical phosphopeptides.

Given that there was no marked preference for specific (patterns of) neutral loss from pHis-containing peptide ions or phosphopeptides containing any other non-canonical residue, observation or lack thereof of specific neutral loss ions was not deemed an appropriate filtering parameter for targeted identification of specific pX-containing peptides. Moreover, our findings that HCD-induced neutral loss is less prevalent from peptides containing pHis and other pX sites compared with pSer- or pThr-containing peptides suggest that the incidence of product ion neutral loss from non-canonical phosphopeptide ions will also be lower. Consequently, the relative abundance of intact phosphosite-determining ions in fragmentation spectra is likely to be proportionally higher for these types of phosphopeptides.

## Identification of non-canonical phosphosites reveals contextual information on phosphoproteins

This new data resource, generated by carefully controlling the pH at which phosphopeptide sample preparation and fractionation are performed prior to LC-MS/MS analysis, provides extensive site and protein information for different non-canonical phosphorylated residues. Consequently, we can, for the first time, determine tentative contextual information for phosphorylation at different pX sites. This information also allows us to compare proteins that are modified by distinct types of phosphorylation events, with certain caveats. A detailed interrogation of the site assignments for pLys and pArg revealed that a number of these phosphosites (66 and 37%, respectively) localised to the peptide C-terminal residue. The catalytic mechanism of trypsin is thought to preclude proteolysis at a phosphorylated Lys/Arg. Consequently, all peptide identifications where pLys/pArg was localised to the C-terminus were removed from subsequent analyses, in case these were examples of false annotation. Analysis of ptmRS scoring suggests that frequent observation of ions at $m/z$ 227.3 and 255.3 (equivalent to Lys or Arg $y_1$ ion + 80 amu) was thus potentially introducing a bias that would not occur for modification of sites at other positions. All other pX data were subject to analysis using Motif-X (Schwartz & Gygi, 2005; Chou & Schwartz, 2011) for consensus enrichment around the site of modification, and DAVID (Huang da et al, 2009a,b) to define enriched gene ontology terms and UniProt identifiers for each non-canonical amino acid. The findings from these analyses are presented below.

### Phosphohistidine

A total of 129 unique sites of pHis were defined at a ptmRS ≥ 0.90 (Fig 3, Datasets EV1 and EV2, Appendix Fig S6, Appendix Table S5). Three of these pHis-containing peptides were generated by chemical phosphorylation of synthetic non-phosphorylated peptides and the tandem mass spectra compared with those acquired in the high-throughput analysis where site localisation was defined as pHis (Appendix Fig S7), increasing confidence in pHis site assignment using the UPAX procedure.

Based on the estimated pAla FLR, 83 pHis sites are thus likely to be true positives, compared with ~99 true-positive pTyr sites identified under the same conditions (Fig 4 and Appendix Table S5). Of the 122 proteins that these pHis sites are found in, 15 (12%) were previously described in the study of phosphohistidine-containing proteins by Hunter and Colleagues (Fuhs et al, 2015) (Table EV1), validating these as cellular pHis-containing proteins. Our new data also allowed us to identify the specific sites of His phosphorylation on these proteins. Interestingly, this list comprises proteins previously immunoprecipitated with both 1-pHis and 3-pHis isomer-specific antibodies, suggesting that our UPAX methodology is capable of identifying proteins containing both pHis isomers. However, UPAX alone does not discriminate pHis isomeric configuration. Importantly, our resource represents the first large-scale *unbiased* investigation striving to define specific sites of pHis in human cell extracts, allowing us to consider contextual information for His phosphorylation in human proteins.

To define a potential consensus motif for His phosphorylation, we used those 37 pHis sites localised with high confidence (ptmRS ≥ 0.99; pAla-estimated FLR of 28%: ~27 true positives) as input data for Motif-X (Schwartz & Gygi, 2005; Chou & Schwartz, 2011). We observed a notable preference for Leu, Val (and to a lesser extent Ile) at +1, relative to pHis, with 21 of the 37 (57%) pHis sites with a ptmRS value ≥ 0.99 matching a pH[L/V/I] consensus (Fig 6A and Appendix Fig S8). Of the 129 unique pHis sites defined in this study (ptmRS value ≥ 0.90), 31% (40) contained this hydrophobic-rich motif around the site of phosphorylation.

To understand the distribution and potential roles of individual phosphosites, we employed the DAVID bioinformatics resource to functionally annotate the different classes of pX-containing proteins. The only enriched category of pHis-containing proteins (ptmRS ≥ 0.90) with an adjusted *P*-value of ≤ 0.05 was "phosphoproteins", with 63 out of 68 proteins being in this category. Relaxing the site localisation confidence threshold to 0.75 ptmRS (with the concomitant increase in FLR to ~51%; Appendix Table S5) additionally revealed enrichment of ubiquitin-conjugated proteins (q-value = 7.67E-05) and those containing coiled-coil regions (q-value = 1.95E-3), or being compositionally biased to contain poly-Ser (q-value = 3.29E-2). Methylated proteins (q-value = 4.08E-2) and those in the cell junction (q-value = 5.00E-2) were also enriched. The relatively small size of this dataset detailing sites of histidine phosphorylation means that it is perhaps unsurprising that the number of significantly enriched categories is lower than that previously described by Hunter and co-workers (Fuhs et al, 2015).

We next used a HEK293T over-expression strategy to validate a novel site of His phosphorylation on the pre-mRNA 3′-end-processing factor FIP1 (FIP1L1; Q6UN15). We obtained confirmatory evidence for a tryptic singly phosphorylated peptide, with site localising fragment ions corresponding to phosphorylation of His 490 on FIP1L1

(Appendix Fig S9). Hinting at the complexity of these analyses, in the same tandem mass spectrum we also observed site-determining ions corresponding to phosphorylation of the adjacent Ser491 residue.

### Phosphoaspartate

Although phosphorylated Asp is recognised as an intermediate in enzyme-catalysed reactions [e.g. P-type ATPases, HAD phosphatases, atypical RIO2 and myosin heavy-chain A kinases and phosphomutases (Ye *et al*, 2010; Attwood *et al*, 2011; Palmgren & Nissen, 2011; Ferreira-Cerca *et al*, 2012)], little is known about pAsp as a covalent protein modification involved in cell signalling. Indeed, searching UniProtKB for 4-aspartylphosphate (or phospho-aspartate) as a modification type in *Homo sapiens* did not identify any proteins known to contain pAsp. While 410 unique pAsp sites were identified at *ptm*RS ≥ 0.90 (342 unique proteins; Fig 3, Datasets EV1 and EV2, Appendix Table S5 and Appendix Fig S10), our pAla decoy search determined a pAsp FLR of 61% at this score cutoff (Fig 4). Statistically therefore, ~161 of these pAsp sites are likely to be novel (*c.f.* ~99 "real" pTyr sites given the pAla-computed FLR).

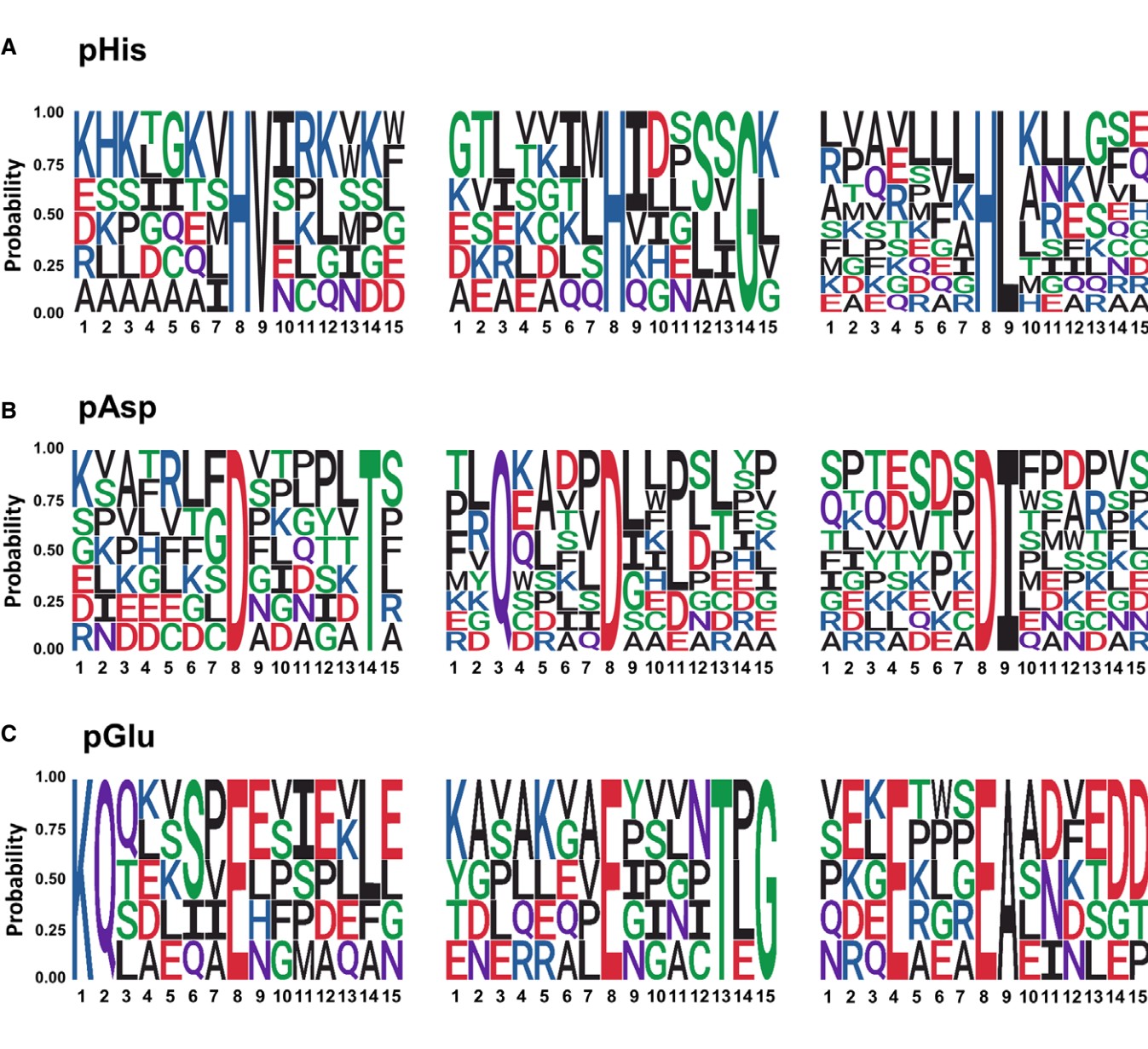

**Figure 6. Motif analysis for pHis-, pAsp- and pGlu-containing peptides.**

A–C   The amino acid sequences surrounding confidently localised sites of (A) pHis, (B) pAsp and (C) pGlu (ptmRS ≥ 0.99) were analysed for sequence enrichment using Motif-X. Depicted are the sequences of the enriched motifs. Additional details are presented in Appendix Figs S8, S11 and S13.

Motif-X analysis of the 99 pAsp sites identified with a *ptm*RS value ≥ 0.99 (pAla-estimated FLR of 57% equates to ~42 true positives) revealed a pD[P/L/I] consensus (Fig 6B and Appendix Fig S11), which was found to be present in 41% of the 99 pAsp phosphopeptide sequences. In agreement with these findings, a pD[P/L/I] consensus was identified in 103 (25%) of all the pAsp peptides with a *ptm*RS ≥ 0.90. A 2.6-fold increase in pAsp-containing peptides with a Q at −5 was also observed for those pAsp sites defined at a *ptm*RS value ≥ 0.99, with a marked preference in this consensus also for I/L at +1 (Fig 6B and Appendix Fig S11).

DAVID bioinformatics analysis (*ptm*RS ≥ 0.90) revealed enrichment of numerous protein classes with an adjusted *P*-value (*q*-value) ≤ 0.05 (Fig 7A), including phosphoproteins (*q*-value = 1.61E-15), acetylated proteins (*q*-value = 1.56E-07), ubiquitin-modified proteins (*q*-value = 2.79E-05) and ATPases (*q*-value = 3.44E-02). Of particular interest is the observation that pAsp-containing proteins are significantly enriched in the nucleus (*q*-value = 1.56E-07), specifically the nucleoplasm (*q*-value = 2.54E-05). The abundance of enriched UniProt keywords and GO terms in categories such as poly(A) RNA binding (74 proteins; *q*-value = 5.21E-07), transcriptional regulation (51 proteins; *q*-value = 1.29E-02) and mRNA splicing (19 proteins; *q*-value = 2.70E-02) suggests a key role for this novel PTM in mammalian transcriptional control.

### Phosphoglutamate

A total of 427 unique pGlu phosphosites were identified across 364 proteins (Fig 3, Datasets EV1 and EV2, Appendix Table S5, Appendix Fig S12). At a *ptm*RS threshold of 0.90, there was a relatively high pAla-derived decoy FLR for the assigned pGlu sites of 79%, meaning that of these, only ~89 pGlu sites are statistically likely to be true positives (Fig 4). Like pAsp, pGlu is not listed in UniProt as a defined modification; thus, there is no curated record of modification of proteins on this residue. In the free-form, pGlu is known as gamma-glutamyl phosphate, which is an intermediate during the glutamine synthase-mediated conversion of glutamate to glutamine. However, very few proteins are established to be modified by phosphorylation on Glu in any organism (Attwood *et al*, 2011).

Unlike pHis and pAsp whose motif analyses predict a preference for hydrophobic residues in close proximity to the site of phosphorylation, analysis of the sequences surrounding the 112 pGlu sites identified at *ptm*RS ≥ 0.99 (pAla-estimated FLR of 69% equating with ~35 true positives in this dataset) revealed two general charge-based motifs: (i) a basic residue-driven consensus, with a preference for Lys at positions −6 or −7, accounting for ~22% of the high-confidence 112 pGlu sites (17% of the sites at 0.90 *ptm*RS); or (ii) a motif biased towards acidic residues (primarily Glu), notably at positions −4 and +3 (Fig 6C and Appendix Fig S13). Indeed, 13% of pGlu peptides contained Glu at −4, while 59% contained one or more acidic amino acids between residues 2 and 6 after the site of phosphorylation (0.90 *ptm*RS).

Similar to our findings for pAsp, DAVID bioinformatics analysis of pGlu-containing proteins (*ptm*RS value ≥ 0.90; Fig 7B) revealed enrichment of proteins in the nucleoplasm (*q*-value = 2.69E-06), poly(A) RNA binding (*q*-value = 1.33E-05), coiled-coil proteins (*q*-value = 2.74E-06), acetylated proteins (*q*-value = 3.04E-05) and ubiquitin-conjugated proteins (*q*-value = 3.59E-04). Unlike pAsp, pGlu proteins were also enriched for chromosome binding proteins

(*q*-value = 4.35E-02), and, in agreement with the Motif-X analysis, proteins containing pGlu were compositionally biased to be Glu-rich (*q*-value = 2.86E-02).

### Phospholysine

Even though phospholysine (pLys) was demonstrated to be present at significant levels in rat liver ~50 years ago (Zetterqvist, 1967; Chen *et al*, 1974, 1977), and proteins with either *in vitro* Lys kinase or pLys phosphatase activity have been identified (Chen *et al*, 1974, 1977; Smith *et al*, 1974; Wong *et al*, 1993), essentially nothing is known about the prevalence or positional distribution of this modification on human proteins. pLys is not currently considered in the controlled vocabulary used in UniProtKB, and consequently, no pLys sites of modification have been recorded in this database. Moreover, following recent studies in yeast, there is growing interest in Lys polyphosphorylation as a non-enzymatic PTM in humans (Bentley-DeSousa & Downey, 2019). In our data, we identify 140 novel sites of non-C-terminal Lys phosphorylation (*ptm*RS ≥ 0.90), mapping to 125 proteins (Fig 3, Datasets EV1 and EV2, Appendix Table S5, Appendix Fig S14). At a pAla decoy-calculated FLR of 41%, this equates to ~82 identified pLys sites statistically being correct (Fig 4).

Of the 45 pLys sites localised with a *ptm*RS score ≥ 0.99 (pAla-estimated FLR of 29% equates to ~32 true positives), a second Lys residue was observed at the position −5 with respect to the site of phosphorylation in 19 (42%) sequences (44% of pLys peptides at 0.90 *ptm*RS) (Fig 8A and Appendix Fig S15). Thirty-three phosphopeptides contained either Gln or Glu at −6 (44% of pLys peptides at 0.90 *ptm*RS). Although Pro was also apparently enriched at +1 with respect to the site of phosphorylation, with 9 out of the 45 pLys sites being following by a Pro (19% of pLys peptides at 0.90 *ptm*RS), we hypothesise that this may be an experimentally induced bias due to the use of trypsin, which is unable to cleavage at Lys (or Arg) residues that are followed by Pro; internal Lys residues are therefore more likely to exist in a LysPro consensus.

DAVID analysis of the 125 pLys-containing proteins (*ptm*RS value ≥ 0.90) identified "phosphoprotein" as the only statistically significantly enriched term (*q*-value = 4.79E-05) for this dataset (Dataset EV1). Increasing peptide identification confidence (1% FDR) additionally revealed enrichment of ubiquitin-conjugated proteins (*q*-value = 6.88E-03).

### Phosphoarginine

Arginine kinase activity has been described in vertebrates. However, the highly basic proteins histone H3 and H4, and myelin basic protein (MBP), all of which are promiscuous protein kinase substrates, are the only Arg kinase substrates reported to date (Smith *et al*, 1976; Wakim & Aswad, 1994; Wakim *et al*, 1995). In our analyses, the number of human pArg sites identified was similar to the numbers mapped for either pTyr or pLys (Fig 3 and Appendix Table S5), and marginally higher than pHis, with 139 unique phosphosites mapped to 116 proteins (Fig 3, Datasets EV1 and EV2, Appendix Table S5, Appendix Fig S16). With a pAla-computed FLR of 23% (lower than the 28% FLR for pTyr), statistically ~107 pArg sites in this dataset are likely to be true positives, higher than the ~99 true-positive pTyr sites (Fig 4). Perhaps not surprisingly given the limited number of proteins identified in our

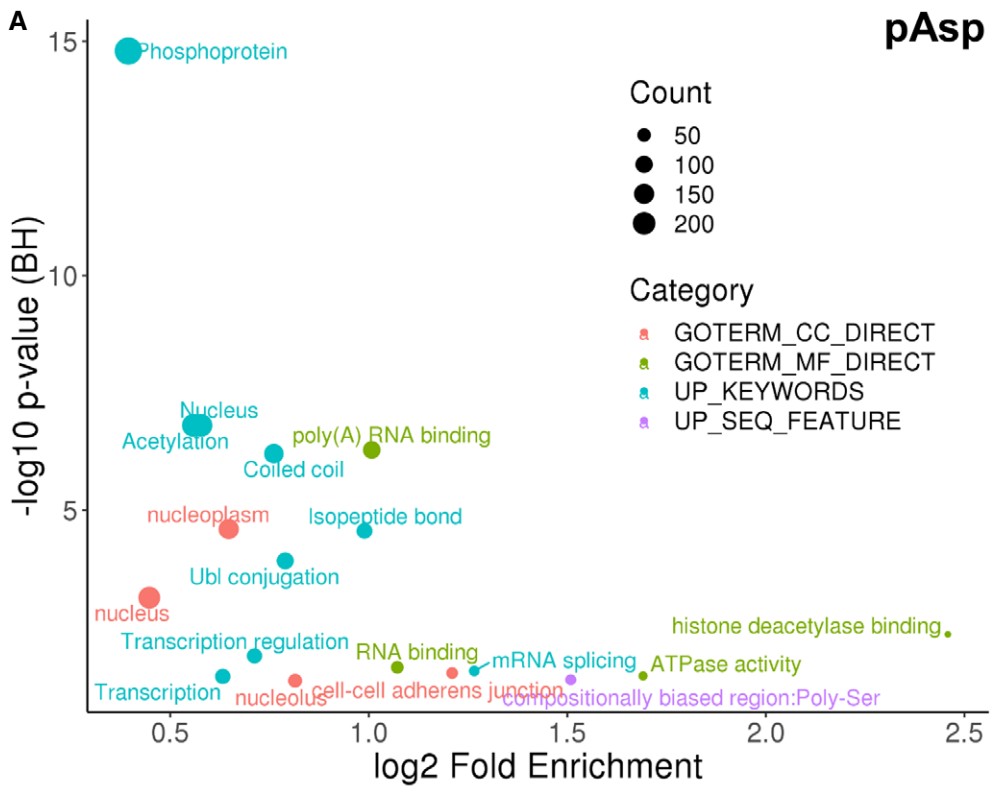

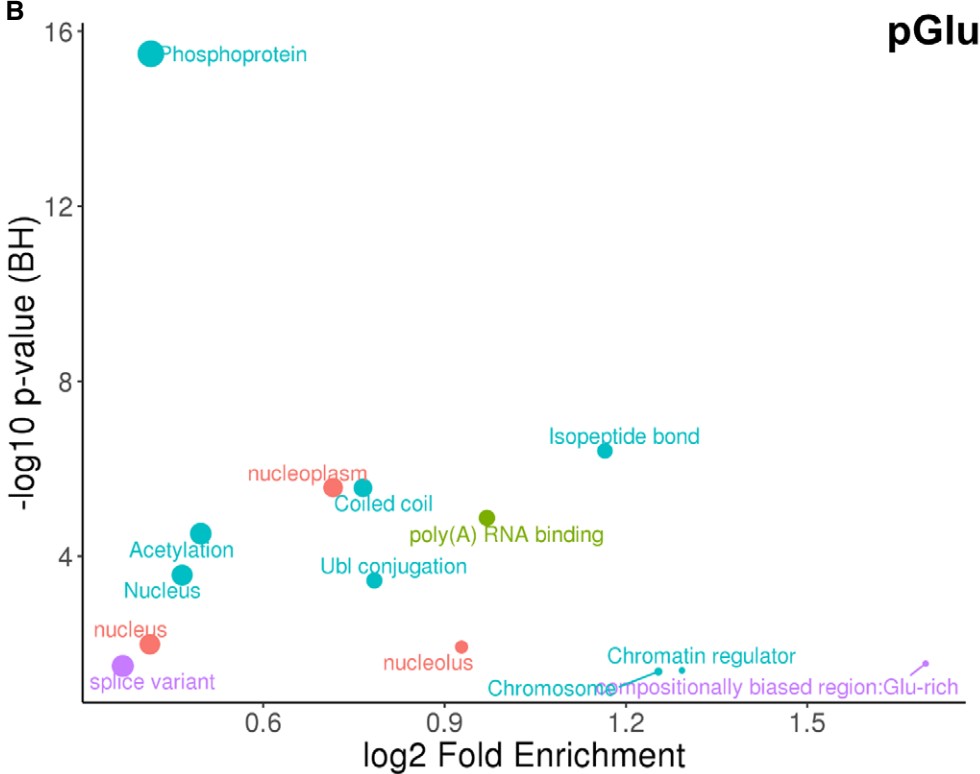

**Figure 7.  Functional analysis of proteins containing either pAsp or pGlu.**

A, B   Functional annotation of the significantly enriched (A) pAsp-containing proteins or (B) pGlu-containing proteins (ptmRS ≥ 0.90) using DAVID (*q*-value < 0.05). BH: Benjamini–Hochberg.

discovery study, previously ascribed pArg-containing histone proteins were not observed. The complexity of combinatorial histone PTMs is such that defining these non-canonical phosphorylation events, alone or in combination with other PTMs, is an important challenge for the future.

Interestingly, DAVID analysis (*ptm*RS value ≥ 0.90; pAla-estimated FLR of 15% equates with ~40 true positives) revealed that in addition to being enriched in phosphoproteins (*q*-value = 3.46E-2), the proportion of proteins compositionally biased to be Arg-rich was also elevated in this dataset (*q*-value = 4.66E-4). However, Arg did not appear particularly prevalent within the 7 residues either side of the phosphorylation site (Fig 8B and Appendix Fig S17). Instead, Motif-X analysis revealed enrichment of Gly at +1, accounting for ~19% of all pArg sites identified. Similar to results observed for

pLys, 10 out of the 47 high-confidence pArg sites (0.99 *ptm*RS) were followed by Pro, possibly due to the same potential trypsin-induced sequence bias discussed for pLys.

### Phosphocysteine

We also identified 55 unique sites of Cys phosphorylation (*ptm*RS ≥ 0.90; Fig 3, Datasets EV1 and EV2 and Appendix Table S5); at a computed pAla decoy-estimated FLR of ~38%, ~34 of these site of identification are thus likely to be true positives (Fig 4). Of these 55 pCys sites, 20 were above the 0.99 *ptm*RS score cut-off applied for motif analysis (at a pAla-estimated FLR of 24%, this equates to ~15 true positives). Eighteen of these contained a hydrophobic residue between −1 and −2, with 13 also possessing a hydrophobic residue at +6 or +7. Five pCys-containing peptides also contained Glu at −7

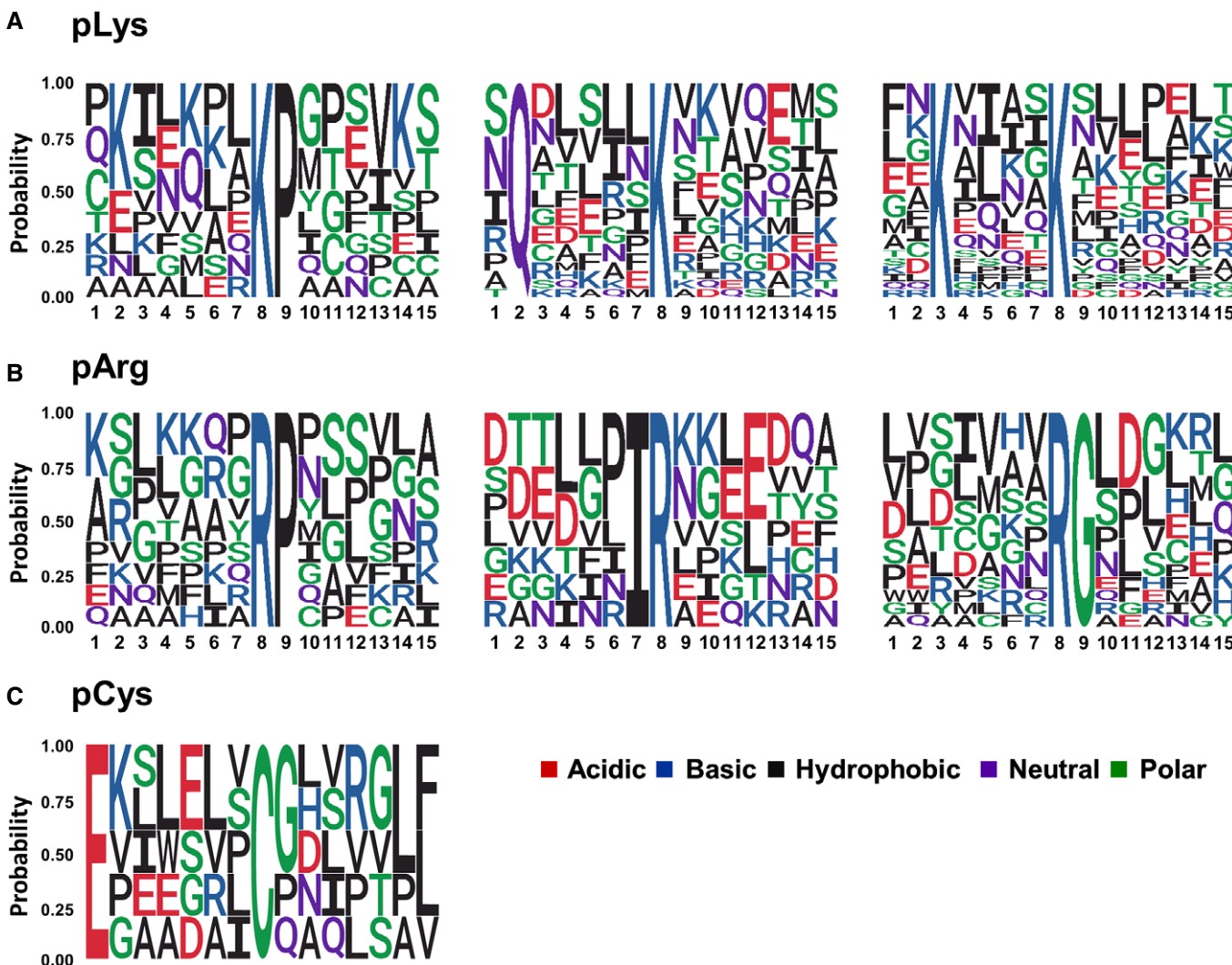

**Figure 8.   Motif analysis for pLys-, pArg- and pCys-containing peptides.**

A–C   The amino acid sequences surrounding confidently localised sites of (A) non-C-terminally localised pLys, (B) non-C-terminally localised pArg and (C) pCys (ptmRS ≥ 0.99) were analysed for sequence enrichment using Motif-X. Depicted are the sequences of the enriched motifs. Additional details are presented in Appendix Figs S15, S17 and S18.

(Fig 8C and Appendix Fig S18). Additional high-confidence pCys site assignments and/or *in vitro* assays will be required to improve confidence in our initial pCys consensus. The small number of pCys-containing proteins prevented identification of enriched protein categories with statistical confidence. However, relaxation of the applied *q*-value threshold (< 0.1) revealed a ~2-fold enrichment in membrane-associated proteins (12 in total; *q*-value = 6.62E-2).

# Discussion

In this resource, we describe an experimental pipeline, termed UPAX, which we have used to identify extensive acid-labile phosphopeptides as well as standard canonical phosphopeptides, from human cell extracts. Altogether, we define ~1,300 novel non-canonical (His, Asp, Glu, Lys, Arg and Cys) phosphosites in HeLa cells cultured under standard conditions, alongside ~3,000 canonical (Ser/Thr/Tyr) phosphorylation sites in the same chromatographically separated SAX fractions. Although the total number of pSer/pThr/pTyr phosphopeptides identified is lower than would be typically observed using traditional phosphopeptide enrichment strategies, e.g. $TiO_2$ or IMAC, we hypothesise that this is because UPAX separates, rather than specifically enriches, phosphopeptides from non-phosphopeptides.

Based on a careful comparison of the number of phosphosites identified using our "all-pX" search strategy with the number of phosphorylation sites identified for the theoretical non-phosphorylatable residue Ala, we show that the commonly applied *ptm*RS score of 0.75 for "class I" phosphosite localisation (Taus *et al*, 2011; Meijer *et al*, 2013; Giansanti *et al*, 2015; Roitinger *et al*, 2015; Lombardi *et al*, 2017) is not acceptable for broad-scale analysis of canonical and non-canonical phosphorylation, since it yields an unacceptably high FLR (Fig 2 and Appendix Tables S5 and S7). To improve confidence in our phosphopeptide datasets, facilitating motif and functional characterisation of the sites and proteins identified, we increased the acceptable site localisation stringency to *ptm*RS ≥ 0.90, further filtering the data to a *ptm*RS value ≥ 0.99 for improved confidence when interrogating pX consensus motifs. Moreover, to address the significant increase in computational search space arising when searching 9 isobaric variable modifications, we used the pAla decoy search to assign confidence levels of individual pX site assignments. Using this approach, we were able to define residue-specific FLR ranging from 13% for pSer to 79% for pGlu, with a pX mean FLR of 29% (5% PSM FDR, 0.90 *ptm*RS), improving to a mean FLR of 16% at *ptm*RS 0.99. These parameters represent a logical compromise between the numbers of individually defined pX sites and site-specific FLR, given the larger than typical overlap between false-positive and false-negative identifications, which is an inevitable consequence of searching large-scale proteomics data for an enhanced number of variable isobaric PTMs. A better understanding of pX-specific phosphopeptide fragmentation patterns combined with more advanced search algorithms will be essential to help overcome this issue as this new field of investigation develops.

To our knowledge, our discovery of widespread non-canonical protein phosphorylation on human proteins catalogued in this resource is entirely novel. Based on the number of sites defined for

each pX residue and the "sham" pAla-computed residue-specific FLR, we conservatively estimate a ratio of 24.6 pSer: 4.3 pThr: 1.0 pTyr: 0.6 pHis: 1.0 pAsp: 0.8 pGlu: 0.8 pLys: 0.9 pArg under standard cell culture conditions. These findings, including our estimation that the number of true-positive sites of pAsp, pGlu, pLys and pArg is broadly equivalent to pTyr in proteins, are unprecedented. In the future, our estimation of the relative ratio of non-canonical to canonical phosphorylation in human proteins will require validation in different systems in order to understand generality, dynamics and cross-species relevance.

Although our studies were performed in the presence and absence of the PHPT1 His phosphatase (Fig 2), no gross increases in the number of pHis (or other pX) sites were identified (Appendix Table S5), suggesting that the pHis phosphatase function of PHPT1 is either redundant or not, in fact, rate-limiting for pHis in human cells after gross suppression of protein levels. Indeed, total phosphopeptide identification rates were found to be lower after siRNA-mediated PHPT1 knockdown, even though equivalent amounts of protein were subjected to UPAX and MS analysis, suggesting both a more general role for PHPT1 in (non)-canonical phosphorylation-mediated signalling and a non-redundant function of PHPT1 as a pHis phosphatase.

Our finding that the non-canonical phosphosites defined in this resource were observed to be significantly enriched in proteins already defined by UniProt as "phosphoproteins" (and to a lesser extent ubiquitin-conjugated proteins) strongly suggests that canonical and non-canonical phosphorylation may work in concert to regulate protein function, perhaps as part of recognised regulatory signalling pathways. Of particular interest is our data relating to pAsp-containing proteins. Not only could we define a hydrophobic residue-driven consensus sequence for phosphorylation (pD[P/L/I]) that accounted for nearly half of all the phosphosites identified, but also we identified significant enrichment of pAsp on proteins involved in (m)RNA binding. The biological significance of this modification will require further investigation.

In terms of mechanism, the introduction of a negatively charged phosphate group adjacent to the basic side chain of His, Arg and Lys residues is of particular interest (Hunter, 2012). The addition of one (or potentially more) phosphate groups (as is the case during polyphosphorylation of Lys and pyrophosphorylation of Ser/Thr) not only changes both the size and polarisation of the side chain, but also reverses the local net charge. By virtue of the near-neutral pKa value for the imidazole proton, His rapidly undergoes catalytically important transitions from a charged to uncharged side chain in many proteins. Consequently, protein phosphorylation on His residues, in addition to well-known examples of enzyme intermediates such as phosphoglycerate mutase, may have a major impact on enzyme-mediated catalysis as well as acting as a conduit to couple (reversible) phosphorylation with modular cellular signalling, in a manner reminiscent of well-characterised pTyr:SH2/PTB domain interactions in vertebrates (McAllister *et al*, 2014). Interrogation of the sites of pHis identified here did not reveal any that were previously classified as enzyme active site intermediates.

The acidic side chains of Arg and Lys are assumed to be fully protonated under physiological conditions. Introduction of a phosphate group with a new directional net negative charge > 1, and an associated "shell" of hydration, could potentially lead to switch-like functional outputs, perhaps including the regulation of catalytic

output or creation of unique binding sites for protein:protein or other biomolecular interactions. The implications of a new mode of competition between phosphate and the myriad of other reversible PTMs that occur on Arg and Lys (most obviously methylation and acetylation) are also noteworthy, and may significantly inform our understanding of combinatorial PTM-driven cell signalling mechanisms and epigenetics, particularly as histones are reported to be subject to phosphorylation on Arg and Lys as well as His (Besant *et al*, 2009).

Although absented from the pX motif and protein enrichment analyses, a large number of pLys and pArg sites were localised by prediction software to the peptide C-terminus, contradicting current understanding of the tryptic catalytic mechanism. We can potentially explain this observation in a number of ways: (i) incorrect site assignment by *ptm*RS based on low intensity and/or a lack of site-determining product ions, and (ii) gas-phase rearrangement of phosphate groups to the C-terminus, either the basic residue side chains or the C-terminal carboxyl group. It is also possible (but less likely) that C-terminal pLys/pArg-containing peptides might arise due to a contaminating protease in the modified porcine trypsin employed.

As exemplified, the high-throughput analysis of canonical and non-canonical phosphopeptides by MS presents a number of new experimental, analytical and computational-associated challenges. For example, the negative free energy of phosphoramidate bond hydrolysis in pHis/pLys/pArg means that peptides containing these non-canonical phosphosites are conceptually more likely to undergo phosphate "scrambling" (phosphosite rearrangement) compared with the relatively low phosphate mobility reported within canonical phosphopeptides (Mischerikow *et al*, 2010). Mechanistic investigation of CID fragmentation of pHis-containing peptides has revealed intramolecular phosphoryl transfer from the pHis moiety to the $\alpha$-carboxyl group to form an acyl phosphate, which is primarily responsible for driving the loss of $H_3PO_4$ ($\Delta 98$ amu) from pHis peptides (Oslund *et al*, 2014). Indeed, gas-phase relocation of phosphate upon CID has been reported for pHis-, pLys- and pArg-containing peptides (Rozman, 2011; Cui & Reid, 2013; Schmidt *et al*, 2013; Bertran-Vicente *et al*, 2014; Gonzalez-Sanchez *et al*, 2014), driven by neutral loss of phosphoric acid. However, although we cannot currently rule out that a proportion of the phosphosites reported in this resource may occur through HCD-mediated phosphate rearrangement, the prevalence of neutral loss from pHis peptides was significantly lower in these HCD spectra than previously described for CID, meaning that the incidence of phosphate scrambling is likely to be much reduced. Indeed, analysis of pHis myoglobin standards did not reveal positional phosphate transfer of any of the five pHis sites during either HCD- or ET-mediated dissociation (ETcaD or EThcD). Nevertheless, it is conceivable that phosphate transfer from N-linked donors to, e.g., pSer-containing peptides could actually mean that the total numbers of pHis, pLys and pArg sites reported may be lower than those present in cells. It is also possible that some C-terminally localised pLys and pArg residues are derived from this acyl phosphate after intramolecular transfer from pHis, pLys or pArg. Future phosphoproteomics studies that seek to define sites of non-canonical (and canonical) phosphorylation will be aided by robust investigations that ascertain the prevalence of HCD (and/or EThcD-mediated) phosphate transfer from all types of phosphorylated amino acids, using *bona fide* synthetic standards, to evaluate the influence of peptide

composition and length. At present however, understanding of the fragmentation mechanisms of non-canonical phosphopeptides is severely hampered by the problems associated with generating suitable chemical standards (Hauser *et al*, 2017).

From an informatics perspective, it is now crucial to develop improved computational strategies to interrogate high-throughput tandem mass spectra for these many isobaric variable modifications. Recent initiatives in "open PTM" search strategies for the identification of different types of undefined PTMs do not come close to adequately addressing the issue of isobaric modifications. The presumed selective phosphorylation of only Ser, Thr and Tyr in human cells means that the interrogation strategies for almost all phosphoproteomics data analysis to date are therefore potentially biased. With this in mind and the (currently limited) evidence of, e.g., gas-phase pArg transfer to Ser, it is conceivable that some of the phosphosites previously ascribed in high-throughput phosphoproteomics studies may, in fact, also be incorrect.

The six non-canonical phosphorylated residues evaluated in this resource (His, Asp, Glu, Arg, Lys and Cys) occur on amino acids that together account for ~28% of all residues found in human proteins. The number of potentially phosphorylatable residues therefore increases to nearly 45% of the total amino acids when canonical Ser, Thr and Tyr residues are also included, meaning that a significant proportion of the human proteome must now be recognised to be potentially subject to phosphorylation. How these distinct types of phosphorylation are regulated, whether enzymatically or via phosphate transfer as established for phosphate transfer from pHis to Asp in two-component signalling systems, represents a major biological challenge for the future. However, the non-random nature of His, Asp, Glu, Lys and Arg phosphorylation site deposition on human proteins defined in our analysis suggests that a myriad of biological roles for these modifications remains to be discovered, which may eventually rival those catalogued for pSer, pThr and pTyr. Validation of the many novel phosphorylation sites revealed here, and enhanced understanding of their physiological roles, will undoubtedly require the development of new tools and methodologies. For example, the production of generic and/or site-specific phospho-antibodies (in addition to those generated for 1-pHis and 3-pHis), alongside chemical genetic strategies for site-specific mimicry or genetic encoding of non-canonical phosphorylated residues, will be critical for mechanistic interpretation of their function. Importantly, our discovery of the widespread nature of novel types of phosphorylation in human cells, and the relative simplicity with which UPAX and MS/MS can be adopted for their analysis in other complex mixtures, argues that the extent and biological relevance of non-canonical phosphorylation events for core and disease cell biology across the kingdoms of life can be revealed quite rapidly. Finally, the genomic annotation of the enzymes that catalyse, promote or hydrolyse non-canonical protein phosphorylation, and subsequent biochemical and cellular analysis to refine our current understanding of eukaryotic signalling remain to be established.

The novel non-canonical phosphorylation sites reported in this resource are likely to represent only the tip of the iceberg; identifying the diverse phosphorylation landscape likely to exist across vertebrate and non-vertebrate organisms is an important challenge for the future. The diversity and prevalence of multiple non-canonical phosphorylation sites raises the question of how they contribute to global cell biology, and whether (as established for pTyr)

the pX marks themselves or the processes that the contextual phosphorylation event controls represent biomarkers, drug targets or anti-targets in disease-associated signalling networks. In this context, it will be interesting to evaluate how non-canonical phosphorylation influences basic biological processes such as the cell cycle, ageing and apoptosis. Moreover, understanding the influence of drugs that modulate signalling pathways, most notably small molecules that were designed to inhibit Ser, Thr and Tyr kinases, might represent a new angle for clinical evaluation of both canonical phosphorylation and non-canonical phosphorylation.

# Materials and Methods

### Preparation of histidine-phosphorylated myoglobin standard

Potassium phosphoramidate was synthesised from phosphoryl chloride and ammonia according to the procedure described by Wei and Matthews (Wei & Matthews, 1991; Hohenester *et al*, 2013; Gonzalez-Sanchez *et al*, 2014). In brief, phosphoryl chloride was reacted with ammonium hydroxide for 15 min on ice producing ammonium hydroxide phosphate, which was added to potassium hydroxide at 50°C for 10 min. Potassium phosphoramidate (PPA) was precipitated with ethanol and collected by vacuum filtration. Equine myoglobin was phosphorylated by dissolution in 1 M aqueous PPA (150 nmol/ml) overnight at room temperature. Phosphorylation was evaluated by intact mass analysis of the resulting phosphorylated protein (5 μM in 20 mM ammonium acetate) by direct infusion via electrospray ionisation (ESI) into a Synapt G2-S*i* mass spectrometer (Waters, UK).

### HeLa cell culture and siRNA knockdown of PHPT1

HeLa cells (ATCC® CCL-2™) were maintained in DMEM (Sigma-Aldrich, Dulbecco's modified Eagle's medium—high glucose, 4,500 mg/l glucose with sodium bicarbonate, without L-glutamine and sodium pyruvate) supplemented with 10% foetal bovine serum, penicillin (100 U/ml) and streptomycin (100 U/ml) at 37°C in 5% $CO_2$. To passage cells, cells were washed with warm PBS (Sigma-Aldrich, phosphate-buffered saline) prior to incubation with 1 ml trypsin (0.05% (v/v)) for 1 min at 37°C. Reaction was quenched with 1 ml supplemented DMEM. All cells are subject to monthly mycoplasma testing. For siRNA knockdown, T75 flasks at ~50% confluency were exchanged to antibiotic-free media (DMEM supplemented with 10% foetal bovine serum). siRNA for PHPT1 (SMARTpool: ON-TARGETplus), Lamin A/C (siGENOME control) and a non-targeting pool (ON-TARGETplus non-targeting pool) were purchased from Dharmacon. For each flask, 1 nM siRNA (1.1 μl of a 20 μM stock prepared in RNAse-free water (Thermo Fisher)) and 40 μl INTERFERin (Polyplus-transfection) were prepared in 4 ml Opti-MEM reduced serum media (Thermo Fisher). After 10-min incubation at room temperature, siRNA mixtures were added to flasks. Cells were incubated for 24 h at 37°C in 5% $CO_2$. For cell lysis, trypsinised cells were centrifuged at 220 *g* for 5 min, washed with PBS and lysed with 100 μl lysis buffer (8 M urea, 50 mM ammonium bicarbonate (AmBic), 1 protease inhibitor tablet (cOmplete Mini EDTA free, Roche) per 10 ml). Lysate was sonicated at low amplitude for 3 × 10 s with a

1-min gap. Protein concentration was determined by the Bradford assay.

### Cell lysis for TiO$_2$-based phosphopeptide enrichment

HeLa cells were pelleted by centrifugation at 220 *g* for 5 min, washed with PBS and then resuspended in 25 mM Ambic with 1× cOmplete Mini EDTA-free protease inhibitor cocktail (Roche) and 1× PhosSTOP (Roche). For lysis, cells were sonicated on ice at low amplitude for 3 × 10 s, then incubated with 0.25% (w/v) RapiGest SF Surfactant (Waters) at 80°C for 10 min. Lysates were centrifuged at 17,000 × *g* for 10 min at 4°C to pellet debris. Protein concentration was determined by the Bradford assay, and 200 μg protein extract was subject to reduction and alkylation, trypsin digestion and RapiGest hydrolysis as previously described (Ferries *et al*, 2017).

### SDS–PAGE and Western blotting

Cell lysates from PHPT1 and non-targeting (NT) siRNA experiments were diluted 1:2 with 2× sample loading buffer (0.06 M Tris–HCl (pH 6.8), 10% (v/v) glycerol, 10% (w/v) SDS, 0.005% (v/v) bromophenol blue, 0.1 M DTT), and normalised for total protein loading. Samples were boiled for 5 min and loaded on a SDS–polyacrylamide gel electrophoresis gel. For Western blotting, proteins were transferred to nitrocellulose, blocked and incubated with the appropriate primary antibody (PHPT1 (Santa Cruz Biotechnology, sc130229, 1:200 dilution); Lamin (Santa Cruz Biotechnology, sc6215, 1:200 dilution); GAPDH (ProteinTech, 60004-1-Ig, CloneNo.: 1E6D9, 1:5,000 dilution)) in blocking buffer overnight at 4°C. Signal was visualised with SuperSignal West Pico PLUS Chemiluminescent Substrate (Thermo Fisher) using X-ray film.

### PEI transfection and immunoprecipitation

Full-length cDNA encoding human FIP1L1 with an N-terminal MYC tag and 3C protease cleavage site immediately after the tag was generated by PCR and ligated into pcDNA3. Purified FIP1L1 plasmid was transiently transfected using a polyethylenimine (PEI, Sigma) procedure in HEK293T cells, cultured in Dulbecco's modified Eagle's medium (DMEM, Gibco) supplemented with 10% foetal bovine serum (FBS, Sigma), 4 mM L-glutamine (Gibco), penicillin and streptomycin (Gibco), and maintained at 37°C in 5% $CO_2$ humidified atmosphere (Longo *et al*, 2013). For transfection, 10-cm tissue culture dishes at 50% confluency were employed. The transfection mixtures, containing a 3:1 ratio of PEI to plasmid DNA (60:20 μg) in nuclease-free water, were pre-incubated for 20 min at room temperature and then added dropwise to the plated cells. Cells were incubated for 24 h, and the growth medium was then supplemented with 4 mM valproic acid (VPA) and cultured for a further 24 h. For cell lysis, trypsinised cells were centrifuged at 220 × *g* for 5 min, washed with PBS and lysed with 100 μl lysis buffer (8 M urea, 50 mM Tris–HCl, pH 7.2, complete protease inhibitor cocktail (EDTA-free, Roche) containing PhosSTOP phosphatase inhibitor cocktail (Roche)). Lysates were sonicated briefly on ice at low amplitude and clarified by centrifugation (16,000 × *g* for 20 min at 4°C), then diluted 10-fold into TBS (final urea concentration 0.8 M). MYC-tagged FIP1L1 was affinity-purified using 50 μl of

pre-equilibrated anti-c MYC-agarose resin (Thermo Fisher) for 2 h at 4°C on a rotating wheel. Beads were pelleted and washed three times in 1 ml TBS buffer. Bound protein was eluted by proteolytic cleavage of the MYC tag using a 1× bead volume of TBS containing 1 µg of His-tagged 3C protease and incubated with gentle agitation for 2 h at 4°C. Supernatant containing the eluted protein was removed and diluted into 25 mM Tris–HCl, pH 8.0.

### In-solution tryptic digestion

Phosphorylated protein standards, histidine-phosphorylated myoglobin (200 µg) and α- and β-casein (Sigma-Aldrich, 100 µg of each) were dissolved in 25 mM ammonium bicarbonate (AmBic) to 2 µg/µl. Protein standards, cell lysates (prepared in lysis buffer as previously described) or immunoprecipitated FIP1L1 were reduced with 3 mM DTT (in 50 mM AmBic) for 20 min at 30°C, and, after cooling, free Cys residues were alkylated with 14 mM iodoacetamide (in 50 mM AmBic) for 30–45 min at room temperature in the dark. The reaction was quenched by addition of DTT to final concentration of 7 mM. The urea concentration in cell lysates was reduced to 2 M by addition of 50 mM AmBic. Proteins were digested using 2% (w/w) Sequencing Grade Modified Trypsin (Promega) at 30°C overnight.

### Titanium dioxide enrichment

$TiO_2$ enrichment of α- and β-casein and histidine-phosphorylated myoglobin peptides (200 pmol) was performed using 200 µl spin tips (Protea Biosciences), with three sets of binding, wash and elution conditions as outlined in Appendix Table S2. Briefly, tips were prepared by addition of binding buffer (200 µl) and centrifuged at 2,000 g for 1 min. Peptides in 200 µl binding buffer were added to the tip, centrifuged, reloaded and centrifuged again. The resulting flow through (unbound material) was collected for analysis. Tips were washed with 200 µl each of binding and wash buffers, with each fractionation collected by centrifugation. Bound peptides were eluted by addition of 2 × 100 µl elution buffer. All fractions were dried by vacuum centrifugation and reconstituted in $H_2O$:ACN (97:3) for LC-MS/MS analysis with the Bruker AmaZon instrument. The non-enriched peptides ("start material") were diluted to a concentration of 500 fmol/µl for LC-MS/MS analysis.

For HeLa-derived peptides, 200 µg of dried peptides was subjected to $TiO_2$ phosphopeptide enrichment according to the procedure outlined previously (Ferries *et al*, 2017), incubating with 1 mg titanium dioxide resin (GL Sciences) for 20 min at 185 × g. For nLC-ESI-MS/MS analysis, peptides were resuspended in 3% (v/v) MeCN and 0.1% (v/v) formic acid and sonicated for 10 min to aid resolubilisation.

### Strong anion exchange (SAX) chromatography

SAX was performed using a Dionex U3000 HPLC instrument equipped with a fraction collector. Peptides from phosphorylated protein standards (25 µg each α- and β-casein and histidine-phosphorylated myoglobin) or digested cell lysate (2 mg) were chromatographed using a PolySAX LP column (PolyLC; 4.6 mm inner diameter (i.d.) × 200 mm, 5 µm particle size, 300 Å) with a binary solvent system of solvent A (20 mM ammonium acetate, 10% ACN)

and solvent B (300 mM triethylammonium phosphate, 10% ACN) at pH 6.0, pH 6.8 or pH 8.0. Solvent was delivered at 1 ml/min according to the following gradient: 5 min at 100% solvent A, then 43 min at 100% solvent B, and then 5 min at 100% solvent B before equilibration to start conditions. Fractions were collected every minute for 48 min, with every 3 pooled, and the volume was reduced by drying under vacuum to give 16 fractions in total.

### C18StageTip desalting

StageTips were prepared with three discs of C18 material in a 200-µl pipette tip. Tips were conditioned by sequential addition of 100 µl methanol, 100 µl $H_2O$:ACN (50:50) and 100 µl $H_2O$, with centrifugation for 2 min at 380 × g to pass the liquid through the tip each time. A portion of each SAX fraction (100 µl) was loaded onto the tip and centrifuged, and then, the flow through was added to the tip and centrifuged again. The tip was washed with 100 µl $H_2O$, and peptides were then eluted by addition of 50 µl $H_2O$:ACN (50:50). Eluents were dried to completion by vacuum centrifugation, then resolubilised in $H_2O$:ACN (97:3) prior to LC-MS/MS analysis.

### LC-MS/MS analysis

#### *AmaZon ETD*
LC-MS/MS analysis of fractions obtained during pHis enrichment optimisation was performed using the AmaZon ETD ion trap (Bruker Daltonics, Bremen, Germany) arranged inline with a nanoACQUITY n-UHPLC system (Waters Ltd., Elstree, UK). Peptides were loaded from an autosampler onto a Symmetry $C_{18}$ trapping column (5 µm packing material, 180 µm × 20 mm) (Waters Ltd., Elstree, UK) at a flow rate of 5 µl/min of solvent A (0.1% (v/v) formic acid in $H_2O$), trapped for 3 min and then resolved on a nanoACQUITY $C_{18}$ analytical column (1.8 µm packing material, 75 µm × 150 mm) (Waters Ltd., Elstree, UK) using a gradient of 97% A, 3% B (0.1% (v/v) formic acid in ACN) to 60% A and 40% B over 60 min at 300 nl/min. The column effluent was introduced into the AmaZon ETD ion trap mass spectrometer via a nano-ESI source with capillary voltage of 2.5 kV. Full-scan ESI-MS spectra were acquired over 150–2,000 *m/z*, with the three most abundant ions being selected for isolation and sequential activation by CID or ETD. A 1-min dynamic exclusion window was incorporated to avoid repeated isolation and fragmentation of the same precursor ion. CID was performed with helium as the target gas, with the MS/MS fragmentation amplitude set at 1.20 V, and ramped from 30 to 300% of the set value. For ETD, peptides were incubated with fluoranthene anions (ICC target 100,000, max ETD reagent accumulation time 10 ms, ETD reaction time 100 ms).

#### *Orbitrap fusion*
nLC-ESI-MS/MS analysis of cell lysate fractions was performed using an Orbitrap Fusion Tribrid mass spectrometer (Thermo Scientific) attached to an UltiMate 3000 nano system (Dionex). Peptides were loaded onto the trapping column (Thermo Scientific, PepMap100, C18, 300 µm × 5 mm), using partial loop injection, for 7 min at a flow rate of 9 µl/min with 2% ACN 0.1% (v/v) TFA and then resolved on an analytical column (Easy-Spray C18, 75 µm × 500 mm, 2 µm bead diameter) using a gradient of 96.2% A (0.1% formic acid in $H_2O$) and 3.8% B (0.1% formic acid in 80:20

ACN:$H_2O$) to 50% B over 90 min at a flow rate of 300 nl/min. A full-scan mass spectrum was acquired over $m/z$ 350–2,000 in the Orbitrap (120K resolution at $m/z$ 200), and data-dependent MS/MS analysis was performed using a top speed approach (cycle time of 3 s), with HCD (collision energy 32%, max injection time 35 ms) and neutral-loss-triggered ($\Delta$98) EThcD (ETD reaction time 50 ms, max ETD reagent injection time 200 ms, supplemental activation energy 25%, max injection time 50 ms) for fragmentation. All product ions were detected in the ion trap (rapid mode).

Immunoprecipitated FIP1L1 tryptic peptides were analysed by nLC-ESI-MS/MS using the Orbitrap Fusion as described above, over a 50-min chromatographic gradient. Peptide ions were subject to HCD nlEThcD, using HCD OT (60K MS1, 30K MS2 at $m/z$ 200).

## Proteomics data analysis

### CompassXport/Mascot

MS output files from the Bruker AmaZon instrument were converted to.mgf files using CompassXport software (Bruker Daltonics) and an in-house script. The resulting.mgf files were searched using the Mascot search algorithm (version 2.6): peptide and MS/MS mass tolerance: 0.6 Da; database: Swiss-Prot (2014.07.21); taxonomy: mammalian; missed cleavages: 2; fixed modifications: carbamidomethyl (C); variable modifications: oxidation (M) and phosphorylation (ST), (Y) and (H). Data were manually inspected using DataAnalysis software (Bruker Daltonics) for the presence of phosphohistidine-containing peptides and to extract peak area values for (phospho)peptides.

### PEAKS 7.5

Label-free quantification of myoglobin peptides in SAX fractions (either with casein or spiked into U2OS lysate) was performed following analysis with the Orbitrap Fusion Tribrid mass spectrometer. Raw files were searched using the PEAKS search engine against either an in-house database created by combining the UniProt Bovine and Equine databases (2016.02.19) or the UniProt human-reviewed database (2015.12.02). Search parameters were as follows: parent mass error tolerance: 10 ppm; fragment mass error tolerance: 0.6 Da; enzyme: trypsin; missed cleavages: 2; fixed modifications: carbamidomethyl (C); variable modifications: oxidation (M), phosphorylation (STY) and (HDERK); max variable PTM per peptide: 4.

### Proteome discoverer

Raw files acquired on the Thermo Fusion mass spectrometer (HCD–neutral-loss-triggered EThcD method) were converted to.mzML using ProteoWizard's msconvert tool in order to perform MS2-level deisotoping. The resulting files were then processed using Proteome Discoverer (PD) v1.4. Data were initially searched by merging all runs into a single search using Mascot (v 2.6) against the reviewed entries of the UniProt reference human proteome (2015.12.02; 20,187 sequences). Parameters were set as follows: MS1 tolerance of 10 ppm; MS2 mass tolerance of 0.6 Da; enzyme specificity set to trypsin with two missed cleavages allowed; fixed modification of carbamidomethylation (C); variable modifications: oxidation (M) and phosphorylation (STY). The results were filtered to a peptide and protein FDR of 1%. This "seed identification search" identified those human proteins present within the sample. The sequences

from the resulting protein matches (both true and decoy) were merged into a.fasta file (7,254 entries), which served as the reduced protein database for the subsequent "all-pX" searches.

The "all-pX" searches were performed using PDv1.4 and Mascot (v2.6) against the identified human protein database created as described above, using the following parameters: MS1 tolerance of 10 ppm; MS2 mass tolerance of 0.6 Da; enzyme specificity set to trypsin with two missed cleavages allowed; variable modifications: carbamidomethylation (C) and phosphorylation (STYHKRDEC). Within PDv1.4, the data were separated according to fragmentation type: HCD or EThcD. HCD and EThcD events were defined using a collision energy (CE) filter (HCD: min CE 0, max CE 34; EThcD: min CE 35, max CE 1,000) to generate two separate.mgf files, which were submitted to Mascot defining the instrument type as either ESI-Quad-TOF for HCD files or CID+ETD for EThcD files. The resulting Thermo Mass Spec Format files were imported into Proteome Discoverer v2.2.

In PDv2.2, scans were selected where the PSM was defined as phosphate-containing group ("phospho" listed in the modifications column), the "Protein groups" column contained "select" or "unambiguous", and the search engine rank was "1". Scans were exported to a new peak information file (.mgf), which was researched using the previously described "all-pX" pipeline but with an additional *ptm*RS phosphosite localisation node and, in the case of pAla search to evaluate FLR, replacing variable phosphorylation of Cys with Ala. The *ptm*RS node was set to ignore neutral loss ions, and the "treat all spectra as EThcD" option was selected for EThcD data. Manual annotation was performed with the help of xiSPEC (Kolbowski *et al*, 2018).

## General bioinformatics analysis

The Python scripting language was used to load and analyse the data outputs from PD. Peptides, including their sequences, *ptm*RS scores and other features were loaded and structured for further analysis.

A PSM FDR filter of either 1 or 5% was applied to all data files (both HCD and EThcD for each pool of SAX fractions). Peptides were further grouped based on whether they contained a phosphosite localised at a score above 1 of 4 given *ptm*RS score thresholds: 0.0 (i.e. all), 0.75, 0.9 and 0.99. Details of the experiment (i.e. NonT or PHPT1), run number (1–3) and fraction number (1–16) were also extracted so that it would be possible to filter or sort on one of these dimensions of this analysis. Such details allowed us to track behaviour of phosphopeptides, particularly during fractionation.

The above description of peptide-based analysis was repeated to produce an analysis that was site-based. In particular, in some cases phosphorylation evidence of a specific protein position could be identified across multiple, possibly overlapping, peptides to lend credibility to the localisation at a protein level. This was done by mapping positions of sites within peptides to the corresponding site in a protein sequence. The site-based analysis also allowed us to view the distributions of probability scores for a particular pX residues, including non-canonical phosphosites to see whether they follow a similar distribution to that of canonical residues. Files were also generated that contained input sequence for motif analysis and protein accessions for functional enrichment analysis.

## Neutral loss characterisation

MS2 spectra corresponding to phosphopeptides were assessed for the presence of 3 neutral loss peaks (Δ80, Δ98 and Δ116 amu) with a mass tolerance of 0.5 Da and an intensity cut-off of 2, 5 or 10% compared to the base peak ion and given a "triplet" score of 0, 1, 2 or 3 depending on the number of neutral loss peaks identified.

## Motif-X analysis

Protein sequences 7 amino acids upstream and 7 amino acids downstream of each unique phosphorylation site were extracted from the UniProt protein sequence database using the Python scripting language. Identical 15-mer sequences were merged together so that the input lists for motif analysis were non-redundant. The R package "rmotifx" was installed from GitHub, and motif analysis tables were generated for all dimensions of our data. To visualise motif discovery, the R package "ggseqlogo" was used to generate the sequence logos, coloured by residue physicochemical properties. To generate sequence logos, all sequences matching a given motif were extracted from the data, as "rmotifx" does not indicate matching sequences, and "ggseqlogo" requires sequences to create a logo.

## Functional enrichment analysis

For functional enrichment analysis, the RDAVIDWebService package for the R scripting language was used. "clusterProfiler" R package was used to create dotplots from these datasets, where *P*-values, enrichment factor, protein count and functional category could be presented.

## Data availability

The mass spectrometry proteomics data (.raw,.mgf and.mzID files) have been deposited to the ProteomeXchange Consortium (http://proteomecentral.proteomexchange.org; Vizcaino *et al*, 2014) via the PRIDE partner repository (Vizcaino *et al*, 2016) with the dataset identifier PXD012188.

Expanded View for this article is available online.

## Acknowledgements
This work was supported by funding from the Biotechnology and Biological Sciences Research Council (BBSRC) to C.E.E. (BB/H007113/1 and BB/M012557/1), to A.R.J. and C.E.E. (BB/R02216X/1) and to A.R.J. (BB/M023818/1, BB/M025705/1 and BB/L005239/1), North West Cancer Research (CR1157, CR1037 and CR1088) to C.E.E. and P.A.E., and a BBSRC DTP PhD studentship to G.H. We would also like to thank Prof. Albert Heck for critical discussions relevant to database searching.

## Author contributions
CEE conceived and designed the study, and performed data analysis; GH, CJC and AEC performed all proteomics sample preparation, and enrichment and mass spectrometry analyses; DPB performed cell culture, and expressed and purified recombinant proteins; GH, CEE, PJB, CJC, AEC, ARJ, AM, SP, AK, DPB and PAE analysed the proteomics data; CEE wrote the manuscript with input from all authors.

## Conflict of interest

The authors declare that they have no conflict of interest.

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
