## [Review Process File · The EMBO Journal]

Strong anion exchange-mediated phosphoproteomics reveals extensive human non-canonical phosphorylation

Gemma Hardman, Simon Perkins, Philip Brownridge, Christopher J. Clarke, Dominic P. Byrne, Amy E. Campbell, Anton Kalyuzhnyy, Ashleigh Myall, Patrick A. Eyers, Andrew R. Jones and Claire E. Eyers.

Review timeline:

Submission date:	6 th February 2019
Editorial Decision:	19 th March 2019
Revision received:	17 th June 2019
Editorial Decision:	19 th July 2019
Revision received:	24 th July 2019
Accepted:	1 st August.2019

Editor: Hartmut Vodermaier

Transaction Report:

1st Editorial Decision

19th March 2019

Thank you again for submitting your manuscript on non-canonical protein phosphorylation to our editorial office, and please excuse the slight delay in getting back to you with a decision after receiving the reports from three expert reviewers and conducting our customary referee cross-consultations. As you will see from the comments copied below, the referees acknowledge the potential interest and significance of your analyses and would in principle consider it suitable for an EMBO J resource article. At the same time, they however all raise a considerable number of substantive issues, concerning analyses, validation, as well as presentation, that would need to be satisfactorily addressed before publication would be warranted.

In light of the overall interest of the subject, I would like to give you the opportunity to respond to and address the reviewers' criticisms by way of a revised manuscript. Should you be able to adequately satisfy the key concerns through additional (re-)analyses and clarifications, we would be happy to consider a revised version further for publication in our 'resource' section. While it may not be essential to include every additional experiment requested in the reports, I would encourage you to get back to me with a proposed outline for answering the referees' points already at an early stage of the revision process, so that we could discuss specific plans for extending the study and key requirements for improvement, as well as concrete revision time frames; this would be particularly important in light of our single-major-revision round policy, which also includes protection from scooping by any competing manuscript appearing elsewhere in press while your work is under consideration by EMBO Press.

REFEREE REPORTS

Referee #1:

In this manuscript, Hardman et al describe a method in which they enrich phosphopeptides from HeLa digest using SAX chromatography. They analyse the later fractions and identify phosphorylation sites on the non-canonical amino acids H, K, R, C, E, D using LC-MS/MS. Remarkably, they identify a similar number of sites for these amino acids as for Y. They test and discuss software problems with site localisation.

This paper is of general interest to a wide readership such as EMBO Journal's as the identification of non-canonical phosphorylation in human cells is adding to the cells' complexity and may open a lot of new biology. It is frustrating that currently no kinases and barely any phosphatases are known for this type of phosphorylation, which limits the ability of the authors to tackle some of the questions genetically. It also actually increases the impact as now people will look more closely for these enzymes. Because this paper potentially has quite some impact, the authors need to be extra careful with their analysis. It is clear to this reviewer that the authors are challenged with a difficult task considering the pH sensitivity of the peptides and the software limitations. The suggestions below may give the reader more confidence and the authors are encouraged to look into them.

Major comments:

1. The biggest concern I have is the use of 5% FDR during the initial search. We all know that going to 5% from the community-accepted 1% is increasing the numbers substantially. What are the numbers with 1% FDR?
2. I am sure that the authors know that they should have used the HCD in the Orbitrap for their MS/MS rather than the ion trap. Have they tried it? Does it lead to an improvement of site localisation?
3. I understand the authors' problems with the search engines. They are usually not performing well with these kind of isobaric modifications. Is there a chance that the authors can search sequentially, thus first non-phospho, then STY and then using the increasingly smaller dataset for the other modifications?
4. The false localisations rates for some of the phosphomods are very, very high, particularly pGlu and pAsp. Would these sites make biological sense? They are already negatively charged to start with. Generally, how do the phosphomods FLR look like when using the $ptmRS > 0.99$? yes, you will lose a lot of peptides but you may get more reliable data.
5. It is clear that standard peptide chemistry cannot be used to make synthetic peptides of any of their hits. However, there are reports of chemistry that were used to make phospho-His, phospho-Lys and others (<https://www.ncbi.nlm.nih.gov/pmc/articles/PMC2143754/> ; <https://pubs.acs.org/doi/full/10.1021/ja507886s?src=recsys>). It would give readers much more confidence in the work if the authors could use some synthetic peptides with these modifications and show that the fragmentation spectra are identical to the ones identified from HeLa digest.
6. I think the figures could do with a make-over.
 - a. There are too many figures.
 - b. Fig 2G is too small.
 - c. Figure 5 can go supplementary.
 - d. The motifs of fig 6-11 could go into one whole page figure. The tables of the motifs which look like they were just copied from Motif-x could be removed by adding number of matches and fold increase (please not with 6 digits!) next to the motifs.

Minor comments:

1. Please add details about the motif-X analysis and general bioinformatics analysis.

Referee #2:

EMBOJ-2018-100647 Eyers

Here the authors developed a new method (UPAX) for enrichment of phosphopeptides under neutral

pH conditions, designed to preserve peptides with phosphate linkages that are labile to acid, including pHis, pLys, pArg, pCys, pAsp and pGlu. Using tryptic peptides from chemically phosphorylated myoglobin, which they showed contains several sites of pHis, they started by showing that conventional phosphopeptide enrichment methods and positive ion mode MS/MS analysis are not suitable for studying non-canonical phosphorylation due to use of the acidic conditions, which result in hydrolysis of phosphoramidate, thiophosphate and mixed phosphor-acid anhydride linkages present in the non-canonical phosphoamino acids. They went on to devise and optimize a phosphopeptide enrichment method that works under neutral conditions, in which strong anion exchange (PolySAX LP) resin is used at pH 6.8, loading in 20 mM ammonium acetate/10% ACN to enrich for total phosphopeptides from a tryptic digest of HeLa cells, followed by elution with 300 mM triethylammonium phosphate/10% ACN. Most phosphopeptides were identified in the later elution fractions, and these fractions were loaded onto a C18 analytical column in 2% ACN/0.1% TFA and then resolved on an analytical column with a gradient of ACN in 0.1% formic (= pH 2.7), and subjected to MS/MS analysis using an Orbitrap Fusion instrument with HCD fragmentation. The resulting spectra were searched using the Mascot algorithm for tryptic peptides in a reduced human proteome database that contained an 80 kDa phosphate signature considering all 9 potential phosphorylatable residues. Extensive efforts were made to reduce the false discovery rate (FDR), and for this purpose the authors devised a new method in which they searched for the presence of a hypothetical phospho-alanine. Setting a criterion of >0.75 statistical probability for the identification of the specific phosphorylation site in an identified phosphopeptide, they authors were able to identify 225 pHis; 626 pLys; 419 pArg; 980 pAsp; 1068 pGlu; and 103 pCys phosphosites, compared with 3636 pSer, 895 pThr; and 236 pTyr phosphosites. They also compared HCD fragmentation with EThcD, observing that the former was more effective, and showing that the phosphate neutral loss pattern using HCD was not diagnostic for phosphorylation of any given residue, and therefore could not be used as a diagnostic. Using a higher >0.90 probability for site identification, the number of pHis sites was reduced to 129, and motif prediction analysis of these sites did not reveal a clear phosphorylation motif, although surrounding residues were enriched in Leu/Ile and Arg/Lys. The number of novel pAsp sites was 161, with again no clear motif but enrichment for Leu and Asp/Glu in surrounding residues; for pGlu the number of high confidence sites was 89, again with enrichment for Leu/Ile/Val and Asp/Glu in surrounding residues; for pLys the number of high confidence sites was 82, with enrichment for Leu/Ile in surrounding residues, and an upstream Lys and a +1 Pro in different predicted motifs; for pArg the number of high confidence sites was 107, with enrichment for Leu/Ile and, like pLys, a +1 Pro in surrounding residues; and for pCys the number of high confidence sites was 34, with a Gly at +1 and an upstream Glu.

Using a newly developed method for phosphopeptide enrichment under neutral conditions, the authors have identified a surprisingly large number of novel sites for pHis, pLys, pArg, pCys, pAsp and pGlu, and this catalogue provides the scientific community with an exciting new resource to study the function of non-canonical phosphorylation events. Their new phosphopeptide enrichment method was extensively validated as being able to correctly identify sites of non-canonical phosphorylation in human cells. However, these findings would undoubtedly be significantly strengthened if the authors could validate some of the novel sites, for example by analyzing an isolated tagged form of the parent protein in question to see if the predicted phosphosite can be identified through site-directed mutagenesis.

1. With regard to histidine phosphorylation, one main issue is that the authors have used the generic term pHis without indicating that there are two isoforms of pHis with distinct properties, and this should be pointed out early in the paper. In this regard, it is unclear whether the UPAX method enriches for both types of peptide equally well. There are several abundant 1-pHis and 3-pHis tryptic peptides that are derived from enzyme intermediates, and therefore it should be possible to determine whether both pHis isoforms can be efficiently recovered during UPAX. They should also indicate that HCD-MS analysis of pHis peptides does not determine whether a peptide contains 1-pHis or 3-pHis.

2. Several of the best characterized pHis sites are derived from enzyme intermediates, and it would be interesting to know how many of the pHis sites that the authors identified are in fact derived from enzyme active sites. Of course, it would be proper to exclude the sequences of such peptides from the Motif-X analysis for sequences preferred for His phosphorylation, because the sequences around these sites are constrained by the conserved active site of these enzymes. In this regard, it is also

unclear whether the apparent enrichment for Leu/Ile in the vicinity of the mapped pHis residues is real, or rather due to selection of this sort of peptide by the SAX column, and/or because these hydrophobic residues protect the pHis residue from hydrolysis under acid conditions or neutral loss. The same question applies to several of the other non-canonical phosphosites, which also have a preponderance of hydrophobic aliphatic amino acids in their vicinity.

3. In terms of the completeness of their survey of pHis sites obtained by their new UPAX method, the authors should indicate how many of the previously reported pHis sites they were able to identify (for instance, pHis peptides, such as that in PGAM1 do not seem to be represented). If bona fide peptides like these are not captured by UPAX, this might mean that the UPAX method exhibits sequence bias, which would limit the ability to do comprehensive surveys (at least for pHis). The authors did a comparison of the pHis sites they identified with regard to whether they were derived from proteins in the list of pHis-containing proteins reported in Fuhs et al. (ref. 12), but since that study did not identify sites of His phosphorylation, it is not clear how many of the identified proteins actually contain pHis - indeed the authors' data provide the first evidence for this.

4. The authors report some peptides with pHis or pLys at the N-terminus, although trypsin cleavage at a Lys/Arg-pHis/pLys bond might be expected to be hindered due the positive charge on Lys/Arg interacting with the neighboring negative charge on the pHis/pLys. In this regard the authors, eliminated peptides where a phosphate was mapped to a C-terminal Lys or Arg (although there still appear to be some peptides of this sort in the list), because trypsin is predicted not to cleave after pLys or pArg. However, they are unable to explain how these phosphopeptides with sites mapped to the C-terminal residue arose during their enrichment/analysis procedure. This raises a question as to whether the pLys and pArg sites they identified at Lys.Pro and Arg.Pro motifs, which also cannot be cleaved by trypsin, might have arisen in the same fashion.

5. Did the authors analyze the extent to which all the identified non-canonical sites are conserved in other vertebrates or lower species, or check structurally, where information is available, to see whether the residues in question are on the surface of the protein and accessible for (enzymatic) phosphorylation?

6. The numbers of pSer, pThr and pTyr sites identified following UPAX enrichment are quite large, but lower than the numbers typically found using conventional phosphoproteomic approaches on HeLa cell samples. Did the authors split a tryptic digest and carry out a comparison of the numbers of canonical phosphosites obtained by the two methods of enrichment? Do the characteristics of the canonical phosphosites from the UPAX analysis differ in any way from those identified using, for instance, TiO₂ enrichment.

Other points:

1. The authors should note that there are at least three reported pHis phosphatases - PHPT1, LHPP and PGAM5 (see ref. 7). In this regard, they could also cite Hindupur et al. (Nature 555:678, 2018), who recently reported that loss of LHPP in liver tumors results in elevated levels of pHis-containing proteins.

2. Page 3 bottom: Why is only pThr referenced here? It would be more interesting to compare all canonical (pSer + pThr + pTyr) phosphorylations versus all non-canonical phosphorylations.

3. Page 4 top: Hydrolysis rate and half-life are also strongly dependent on temperature, and so these values will change over the range from 40°C to 90°C. For example, at pH 4 at 90°C, 100% of signal was degraded in 10 min. The temperature used is indicated in the Materials and Methods (25°C), but it would be important to consider this aspect within the main text.

4. Page 4: What do the authors know about the 1/3-pHis identity of the His sites phosphorylated chemically in myoglobin with phosphoramidate? Is it clear that myoglobin Lys residues were not also phosphorylated under these conditions?

5. Page 4 bottom: Does this mean that the elution is time dependent or volume dependent? Does a longer period (lower flow rate) with TEA elute phosphopeptides in earlier fractions? Are the peptides identified in early fractions only non-phosphorylated peptides from incomplete chemical

phosphorylation or did they lose their phosphate during the whole process by hydrolysis as shown in Fig. EV3? If the phosphopeptides are eluted in a time-dependent manner, then the fact that the later fractions that eluted at an increased pH contained all the phosphopeptides might mean that increased pH reduces the hydrolysis.

6. Page 5: The authors should discuss the potential issues with the LC-MS/MS loading and resolving buffers being rather acidic, and how this might affect the recovery of phosphopeptides with non-canonical phosphoamino acids, due to the potential for phosphate loss during loading and chromatographic resolution at pH ~2.5.

7. Page 5: It appears that the pH gradient elution using TEA helped to separate unphosphorylated peptides from phosphopeptides in the later fractions. This suggests that nearly no phosphates were lost during the ionization process at pH 2 and higher temperature when injected into the instrument, since 90-100% of peptides phosphorylated were obtained in these fractions. How do the authors explain this surprising stability? Is this a result of short loading/injection times (c.f. Potel et al. ref. 26)?

8. Page 7: The strategy is interesting, but it is not really clear whether the final list of proteins is from pSer/Thr/Tyr peptide site identification by MS/MS or phosphopeptide identification by MS. What do the authors mean by "first pass search" here? It will be totally different as the mass of a tryptic peptide + phosphate will not change regardless of the position of the phosphorylated residue, but the theoretical mass of a fragmented ion pS/T/Y (MS/MS) will change the m/z ratio as a function of the phosphoresidue and increase the combinations that can bias the final option even with a high resolution instrument.

9. Page 7: Is there a final listing for each unique phosphoresidue? For example, do the 225 unique pHis sites come from the 289 His sites in the list of phosphopeptides >0.75 (c.f. EV1) once the low probability ones are excluded that were identified as containing another phosphoresidue at >0.75 on the same peptide?

10. Page 8: Did the authors use a stochastic model with all the potential phosphoresidues from their "limited human database" or did they consider the full human proteome? (See comments on Material and Methods section)

11. Page 17: In line 7 and in the second paragraph of the section on "Phosphoarginine", the authors say pLys, when presumably they mean pArg?

12. Page 18: Do any of the pCys sites correspond to the active site peptides from proteins like PTPs, which are known to have a pCys intermediate?

13. Page 21: Was the 28% number derived from the limited human proteome database used? Otherwise these 6 residues should represent more than 60% of the entire human proteome sequence.

Materials and Methods:

1. Page 7: As specified by the supplier, hydroxyapatite should never be centrifuged, because it explodes the resin beads and strongly decreases efficiency, especially if an acid condition is used.

2. Page 7: Hydroxyapatite acts as an affinity resin, and so a longer incubation as with antibody binding is not necessary.

3. Page 7: The idea of using pAla and normalizing to the numbers of each amino acid is innovative, but the rationale or relative frequencies will be different for every peptide according to the different sequence, meaning that in principle every peptide should be normalized on its own sequence. Even if every peptide is considered independently, if one imagines a scenario with the same residue at the C-terminus and N-terminus and thus with at least a twofold higher chance to be phosphorylated than any unique site between, there is still zero risk to position the phosphate at the opposite position

with the ion series as it will be necessarily fragmented. It is for this reason that small peptides (less than 6-7 amino acid) are usually not considered

4. Page 8: The chance of mislocalizing the position of phosphate after MS/MS is more likely due to the space between two potential residues. So if the frequencies of residues are considered for mathematical correlation, a risk factor of distance between these residue should be also integrated. Yet, another aspect is that His is the only residue that can generate two phosphoforms (1- and 3-pHis) - does this mean that the FLR should be divided by two when this will likely depend on the steric hindrance and the tertiary structure of the protein.

5. Page 8: Is the C-terminal phosphoresidue normalized by the non-C-terminal calculation or are they not normalized at all because the Ala can never be at the C-terminal position?

6. Page 9: How do the authors explain that, considering the normalization, there is an increase in pHis signal for pH 4/pH 6 at $t = 15$ min and $t = 1$ h compared to $t = 0$? If this is from free phosphoramidate, then the chemical phosphorylation would still be ongoing, meaning that one is not defining the degradation, but rather turnover.

7. Page 10: Does the 33% increase in non-phosphorylated peptides after neutral loss correspond to 100% of phosphorylated peptides or 1%? What was the original ratio of phosphorylated versus non-phosphorylated peptides in the load? Were there some pHis-containing peptides that were eluted from the resin, but potentially lost their phosphate subsequently by hydrolysis?

8. Page 18: Were there any eluted His-containing peptides lacking phosphate identified that corresponded to identified pHis peptides?

Minor points: 1. Abstract: Cys phosphorylation should be mentioned in the Abstract.

2. Keywords: Why choose to specify phosphoaspartate as a key word and not any of the other non-canonical phosphoaminoacids?

Referee #3:

Comments

The manuscript presented by Hardman et al. described their discovery of endogenous phosphorylation post translational modification on a wide range of residues, including His, Lys, Arg, Asp, Glu and Cys. The authors employed a Strong Anion Exchange based separation approach, termed UPAX, to preserve labile phosphorylation modification on these non-conventional sites and used high resolution LC-MS/MS to identify a novel set of phosphorylation sites. To address inflated false site discovery problem, the authors developed a search strategy to allow estimation of false localization rate. Overall this manuscript is a valuable addition to the phospho-proteomics field by introducing a novel approach to discover unconventional phosphorylation sites. A few key questions are well addressed, such as false discovery rate and motifs. That's being said, I have a few additional questions and comments to the authors.

Major comments:

1. A main experiment missing from this manuscript is negative control. Although the authors have an attempt to knock down histidine phosphatase, no apparent difference in the number of pHis is observed. An alternative control could be acid-treated or heat-treated peptide digests.

Isotopic/isobaric labeling may be applied to the acid-treated and untreated peptides, combined and separated on UPAX. A true non-canonical phosphosite should demonstrate decreased quantitative signal due to hydrolysis. This would provide a strong support to the identity of reported sites.

2. What is the identification rate of unconventional phosphosites on a TiO₂ enriched global phospho data set when modification is permitted on H/K/R/D/E/C as well as S/T/Y during mascot database search? How is it compare to UPAX approach?

3. It's not entirely clear to me that how three biological replicates are summarized, i.e. only the sites

identified in all three are retained in the final set, or observation in any replicates are included in the final set?

4. High frequency of c-terminal lysine phosphorylation is a concern to me. Is it the true false positive rate? Such frequency greatly exceeds pAla estimated FLR rate. I'd like to see more analysis or validation on this subset, rather than just excluding from further analysis. For example, can these sites still be identified when an enzyme with different specificity is used?

Minor comments:

1. Figure 1 A-C: how many peptides are summarized here? Please indicate it in the figure legend.
2. Figure 2 A indicates HeLa cells 24h but figure legend shows 48h. Please confirm.
3. Page 17: ~107 pLys sites Should be pArg sites.

1st Revision - authors' response

17th June 2019

Referee #1:

This paper is of general interest to a wide readership such as EMBO Journal's as the identification of non-canonical phosphorylation in human cells is adding to the cells' complexity and may open a lot of new biology. It is frustrating that currently no kinases and barely any phosphatases are known for this type of phosphorylation, which limits the ability of the authors to tackle some of the questions genetically. It also actually increases the impact as now people will look more closely for these enzymes. Because this paper potentially has quite some impact, the authors need to be extra careful with their analysis. It is clear to this reviewer that the authors are challenged with a difficult task considering the pH sensitivity of the peptides and the software limitations. The suggestions below may give the reader more confidence and the authors are encouraged to look into them.

Major comments:

1. The biggest concern I have is the use of 5% FDR during the initial search. We all know that going to 5% from the community-accepted 1% is increasing the numbers substantially. What are the numbers with 1% FDR?

Additional information, including an expanded view Figure (Figure EV4) and supplementary table, has now been added to the revised manuscript that incorporates these data. As expected, the number of unique phosphopeptides identified decreases at a 1% FDR, with the decrease in the number of unique phosphopeptides identified being maximal for those containing pCys (at 58%). However, the difference across all pX sites was not as large as might be expected (Reviewer Table 1).

ptmRS>0.9	pSer	pThr	pTyr	pHis	pAsp	pGlu	pLys	pArg	pCys	pAla
Total pX peptides	8.1%	21.8%	32.1%	44.2%	21.2%	27.7%	56.2%	42.4%	51.4%	20.9%
Unique pX sites	13.5%	34.8%	42.8%	45.7%	22.9%	34.0%	48.6%	51.8%	58.2%	32.3%

Reviewer Table 1: *Percentage decrease in the number of phosphopeptides and unique phosphorylation sites identified upon application of a 1% FDR cut-off compared with applying a 5% FDR, at a 0.90 ptmRS score.*

Importantly, there is little overall difference in the pAla-computed FLRs with a reduction in FDR from 5% to 1%: pSer decreases to 10% from 13% (*ptmRS* 0.90), while there is actually an increase of FLR for pTyr and pHis from 28% to 31% and 36% to 42% respectively (new Appendix Table S6). This additional data is now included as Supp. Fig S4 and the implications of FDR and *ptmRS* selection of phosphopeptide numbers and pAla-computed FLR are discussed more extensively in the text.

2. I am sure that the authors know that they should have used the HCD in the Orbitrap for their MS/MS rather than the ion trap. Have they tried it? Does it lead to an improvement of site localisation?

As we have previously shown (Ferries *et al.*, JPR 2017), high resolution Orbitrap-based MS analysis (HCD OT) does indeed serve to increase the accuracy of phosphosite localisation compared with low resolution ion trap-based MS analysis (HCD IT) (from 87% to 92% in our benchmark study of a synthetic phosphopeptide library set). To evaluate the effect of HCD OT on phosphosite localisation confidence here, we analysed one of the latter SAX fractions (containing a high proportion of phosphopeptides) using both HCD OT and our current HCD IT method. Although we were unable to evaluate phosphosite localisation accuracy (given a lack of *bona fide* pX-containing standards) we could consider the number of unique phosphopeptides and phosphosites obtained. At a *ptmRS* cut-off value of either 0.90 or 0.75 we identified 30% or 44% fewer unique phosphorylation sites respectively when using an HCD OT regime in a single pX-dense SAX fraction. The reviewer is correct in that there will likely be significant benefits of applying different MS acquisition parameters, including high resolution data acquisition, differential fragmentation regimes including ETD and possibly UVPD, in combination with alternate proteases, to improve confidence of pX site identification. These various approaches are currently being evaluated.

3. I understand the authors' problems with the search engines. They are usually not performing well with these kind of isobaric modifications. Is there a chance that the authors can search sequentially, thus first non-phospho, then STY and then using the increasingly smaller dataset for the other modifications?

Although from a search space perspective, this type of iterative searching would be preferable, interrogation of this dataset as suggested by the reviewer is not feasible, precisely because the modification (phosphorylation of S/T/Y or phosphorylation of a non-canonical residue) is isobaric. Sequential searching for phosphorylation in this manner leads to mis-assignment (as discussed in our BioRxiv preprint 20280, Hardman *et al* 2017). A similar type of phosphoresidue-selective searching as suggested by the reviewer was performed in the data presented in our related BioRxiv pre-print, where different non-canonical residues were considered, in addition to pSer/Thr/Tyr, at a time. Our analysis showed that ~23% of the 3237 total peptide spectrum matches (PSMs) containing non-canonical phosphorylation across all searches were derived from the same MS/MS scan, with a different pX site defined at a 1% FLR for the same peptide sequence, strongly indicative of extensive mis-assignment of phosphorylation site.

To exemplify: consider peptide sequence ADDLDFETGDAGASATFPMQCSALR from Eukaryotic translation initiation factor 5A-1 (EIF5A); the Asp at position 10 was confidently localised (0.9995 *ptmRS*) as the site of phosphorylation. If pAsp had not been considered during searching of the tandem mass spectra for this peptide, there is a high likelihood that either the Thr at position 8, or the Ser at position 14 would have been defined as the site of modification in the absence of the ability of the search engine to consider the alternative pAsp residue.

4. The false localisations rates for some of the phosphomods are very, very high, particularly pGlu and pAsp. Would these sites make biologically sense? They are already negatively charged to start with.

Phosphorylation of Asp and Glu on endogenous proteins is already documented (see *e.g.* review by Attwood et al *Amino Acids* (2011) DOI 10.1007/s00726-010-0738-5) and proteins with kinase (and phosphatase) activity towards aspartate and glutamate have been identified in a limited number of organisms. Although evidence of proteins containing pGlu currently appears to be limited to prothymosin α and collagen, there is strong evidence of biological occurrence of pAsp, both as part of two-component signalling systems in eukaryotes, fungi and plants, and also in mammalian cells, primarily as an enzymatic intermediates in the active site of proteins such as the P-type ATPases. Consequently, while we acknowledge (and discuss extensively in the manuscript) that the false localisation rates for pAsp and pGlu are relatively high, phosphorylation of these negatively charged residues on proteins is a validated biological phenomenon. Whether the presence of this, more bulky, negatively charged phosphate moiety serves to directly facilitate conformational change of a protein (or interactions with other macromolecules), or whether these labile phosphate anhydrides serve as energy stores to mediate some other phenomenon (as proposed by *e.g.* Stock and Da, 2000 *Current Biology*), will need to be determined on a case-by-case basis.

Generally, how do the phosphomods FLR look like when using the $ptmRS > 0.99$? yes, you will lose a lot of peptides but you may get more reliable data.

The comparison of the number of unique phosphosites at a 0.99 *ptmRS* cut-off as opposed to 0.90 or 0.75, is presented in the original version of the manuscript in Fig. 3A, Fig 3B, Expanded view Dataset 1, Appendix Table S5, S6 and is discussed extensively in the main text. The new Expanded view dataset 2, now includes details of phosphorylation sites in a manner that is more readily searchable. The reviewer is correct that some phosphopeptides are lost at this high stringency *ptmRS* value (irrespective of the type of phosphorylation event), with a concurrent reduction in the pAla-computed FLR as *ptmRS* cut off increases. Indeed, because of this, we have not used the standard 0.75 *ptmRS* score cut-off typically applied to generate ‘Type I phosphopeptides’, and elected to use only those phosphopeptides that met the 0.99 *ptmRS* site localisation criteria for consideration when calculating motif conservation around each of the non-canonical phosphorylated residues.

5. It is clear that standard peptide chemistry cannot be used to make synthetic peptides of any of their hits. However, there are reports of chemistry that were used to make phospho-His, phospho-Lys and others (<https://www.ncbi.nlm.nih.gov/pmc/articles/PMC2143754/> ; <https://pubs.acs.org/doi/full/10.1021/ja507886s?src=recsys>). It would give readers much more confidence in the work if the authors could use some synthetic peptides with these modifications and show that the fragmentation spectra are identical to the ones identified from HeLa digest.

This is an important point. Using phosphoramidate-mediated His phosphorylation procedures that we employed to generate pHis-containing Myo, we chemically synthesised 8 pHis-containing peptides that were identified in our original UPAX dataset. Three peptides could not be re-solubilised after lyophilisation, and we were unable to generate stable phosphohistidine-containing versions of two others. However, we were able to validate the site of phosphorylation on three synthetic pHis peptides by HCD OT, comparing the HCD IT data for these synthetic peptides with those previously obtained in the original study (Appendix Fig S7). Unfortunately, we do not have the chemical synthesis

capabilities to generate pLys standards and have been unable to find any published protocol for synthesis of pAsp or pGlu-containing peptides.

6. I think the figures could do with a make-over.

a. There are too many figures.

b. Fig 2G is too small.

c. Figure 5 can go supplementary.

d. The motifs of fig 6-11 could go into one whole page figure. The tables of the motifs which look like they were just copied from Motif-x could be removed by adding number of matches and fold increase (please not with 6 digits!) next to the motifs.

We agree with the reviewer that some of our original figures were a little complex. Throughout the manuscript, we have simplified these where possible, combining the main findings from the Motif-X analysis into two figures, the GO analysis to a separate figure and the Motif-X tables (amended!) to the supplementary material. We have also increased both the font and size of key parts of the figures as suggested. With regard to Figure 5, this data is crucial to the discussion on phosphopeptide neutral loss as a function of likely pX site and as such cannot really be moved to the supplementary, being critical to the flow of the manuscript.

Minor comments:

1. Please add details about the motif-X analysis and general bioinformatics analysis.

Additional text describing these processes have now been added to the main Methods section.

Referee #2: (Comments condensed to those points needing to be addressed):

Using a newly developed method for phosphopeptide enrichment under neutral conditions, the authors have identified a surprisingly large number of novel sites for pHis, pLys, pArg, pCys, pAsp and pGlu, and this catalogue provides the scientific community with an exciting new resource to study the function of non-canonical phosphorylation events. Their new phosphopeptide enrichment method was extensively validated as being able to correctly identify sites of non-canonical phosphorylation in human cells. However, these findings would undoubtedly be significantly strengthened if the authors could validate some of the novel sites, for example by analyzing an isolated tagged form of the parent protein in question to see if the predicted phosphosite can be identified through site-directed mutagenesis.

The reviewer makes a valid point. To this end we over-expressed MYC-tagged versions of 5 proteins identified as containing site of non-canonical phosphorylation in HEK293T cells and subjected them to LC-MS/MS analysis for validation (as described in the updated Methods section). Of these 5, we were unable to unambiguously define the phosphorylation site as occurring on a non-canonical residue, given that all were in close proximity to another potential site of modification and in many cases there was evidence of chimeric tandem mass spectra, i.e. singly phosphopeptide peptides containing one of two potential sites of modification were co-isolated and subject to tandem MS, meaning that product ions derived from both phosphoisomers were present in the spectra. As an example we present the EThcD mass spectrum of the peptide from Pre-mRNA 3'-end-processing factor FIP1 (FIP1L1) which we identified as containing pHis, for which we show evidence of product ions associated with the likely presence of both pHis at position 2 in the peptide (position 490 on FIP1L1) and pSer at position 3. This has been included in the main text and the spectrum is now presented in Appendix Fig. S9. Clearly more work is needed for

unambiguous validation of individual pX sites, which will likely require site-by-site optimisation of fragmentation strategy and protease (to generate peptides large enough, but not too large, for analysis by *e.g.* LC-MS/MS with an ETD fragmentation regime).

1. With regard to histidine phosphorylation, one main issue is that the authors have used the generic term pHis without indicating that there are two isoforms of pHis with distinct properties, and this should be pointed out early in the paper. In this regard, it is unclear whether the UPAX method enriches for both types of peptide equally well. There are several abundant 1-pHis and 3-pHis tryptic peptides that are derived from enzyme intermediates, and therefore it should be possible to determine whether both pHis isoforms can be efficiently recovered during UPAX. They should also indicate that HCD-MS analysis of pHis peptides does not determine whether a peptide contains 1-pHis or 3-pHis.

As detailed in Table EV1 where we compare our pHis dataset with proteins previously identified as being quantitatively enriched with either the 1-pHis or 3-pHis mAbs by Fuhs et al., we can confirm that our method permits us to identify both pHis isomers. We have now included an additional sentence in the pHis section about lack of discrimination using our UPAX strategy, and that both isomers are likely to be observed.

2. Several of the best characterized pHis sites are derived from enzyme intermediates, and it would be interesting to know how many of the pHis sites that the authors identified are in fact derived from enzyme active sites.

We have evaluated our identified pHis phosphosites for potential known roles as pHis enzyme intermediates, such as NME1/NDPKA (based on UniProt annotations) but were not able to identify any overlap. However, as stated in the review, this resource defines methods and a starting point from which to explore pHis and other labile sites of modification, so we anticipate that these will be identified as coverage of the non-canonical phosphoproteome improves.

Of course, it would be proper to exclude the sequences of such peptides from the Motif-X analysis for sequences preferred for His phosphorylation, because the sequences around these sites are constrained by the conserved active site of these enzymes.

This is not pertinent given our response to the previous statement. However, it is worth stating that we are not looking to predict substrate motifs for enzymes, just the likelihood of a pHis within any amino acid consensus, which would naturally include enzyme intermediates *e.g.* NME1/NDPKA.

In this regard, it is also unclear whether the apparent enrichment for Leu/Ile in the vicinity of the mapped pHis residues is real, or rather due to selection of this sort of peptide by the SAX column, and/or because these hydrophobic residues protect the pHis residue from hydrolysis under acid conditions or neutral loss. The same question applies to several of the other non-canonical phosphosites, which also have a preponderance of hydrophobic aliphatic amino acids in their vicinity.

We are currently investigating the consensus motifs surrounding pX sites when UPAX is applied using alternative enzymes to trypsin to establish if different consensus motifs for pX phosphorylation are additionally identified. However, this will be an extensive study and there is not scope to include these investigations in this current manuscript. Given the SAX medium, there is a possibility of marginal enrichment based on hydrophobicity, although this should be largely negated by the inclusion of MeCN in the chromatography buffer. In support of this, there was no consensus for hydrophobic residues for the pSer peptides in our dataset, confirming no overt SAX-based bias for Leu/Ile/Val residues

within phosphopeptide motifs. It is also worth noting that current procedures for phosphopeptide enrichment (TiO₂ and IMAC) are themselves biased towards highly acidic phosphopeptides and indeed, there is a differential preference for these two chromatographic media for singly and multiple phosphorylated peptides respectively.

3. In terms of the completeness of their survey of pHis sites obtained by their new UPAX method, the authors should indicate how many of the previously reported pHis sites they were able to identify (for instance, pHis peptides, such as that in PGAM1 do not seem to be represented).

Table EV1 shows a direct comparison of those proteins identified in the most complete human pHis proteome to date, published by Fuhs *et al.* 2015. Of the 122 pHis proteins that we identified, 15 of these (12%) have previously been identified. The potential limitations of UPAX, in terms of depth of phosphoproteome coverage is discussed in the manuscript.

4. The authors report some peptides with pHis or pLys at the N-terminus, although trypsin cleavage at a Lys/Arg-pHis/pLys bond might be expected to be hindered due the positive charge on Lys/Arg interacting with the neighboring negative charge on the pHis/pLys.

We have performed a systematic evaluation of data from the PeptideAtlas phosphoproteomics build, in which a large number of publicly available data sets have been re-processed with a standardised pipeline, and high-confidence false localisation rate. We examined the position of the pS, pT and pY sites along the peptide chain. For pS and pT there is in fact a strong enrichment for pS and pT to be the N-terminal amino acid – in the case of pS, this is in fact the most common position on a peptide where a phosphorylation site can occur. We believe this is due to the relatively common basophilic [K/R]S kinase recognition motif. For pY, there is no strong enrichment, but phosphorylation at the N-terminal most residue seems as likely as any other position within a peptide to be phosphorylated. Overall, there is no evidence that phosphorylation on S, T, Y inhibits tryptic cleavage when the preceding amino acid is K or R. We feel it would distract from the main message of the paper to include this data, although we are preparing a global analysis of this data for a separate manuscript.

In this regard the authors, eliminated peptides where a phosphate was mapped to a C-terminal Lys or Arg (although there still appear to be some peptides of this sort in the list), because trypsin is predicted not to cleave after pLys or pArg. However, they are unable to explain how these phosphopeptides with sites mapped to the C-terminal residue arose during their enrichment/analysis procedure. This raises a question as to whether the pLys and pArg sites they identified at Lys.Pro and Arg.Pro motifs, which also cannot be cleaved by trypsin, might have arisen in the same fashion.

We thank the reviewer for highlighting the remaining C-terminal pLys peptides in the EV Dataset 1 – these have now been removed.

Peptides containing pLys or pArg at the C-terminal position of the peptide were specifically excluded due to the nature of the *ptmRS* scoring algorithm; scoring at the C-terminal position can be (uniquely) biased by the presence of y1+80 fragment ions. This is a potential flaw in the *ptmRS* algorithm, which is not an issue in almost any other usage of the tool, since it is rare to search for modifications that could occur on C-terminal lysine or arginine residues. Regarding the identification of pLysPro and pArgPro sequences, these may indeed be a consequence of using trypsin as the enzyme for these analysis. Ongoing studies are investigating the effect of using alternate proteases on the occurrence and relative localisation of pLys and pArg.

5. *Did the authors analyze the extent to which all the identified non-canonical sites are conserved in other vertebrates or lower species, or check structurally, where information is available, to see whether the residues in question are on the surface of the protein and accessible for (enzymatic) phosphorylation?*

We have begun to explore the conservation of modified amino acids in other species, but we respectfully wish to exclude the analysis from this manuscript. The most challenging aspect of this analysis is that it is currently unknown as to what levels of conservation might be expected for these amino acids. Even for pSTY sites, there are mixed results in the literature as to whether they are more or less conserved than non-phosphorylated sites between diverse species, and this is hampered by the fact that it is hard to create a “background” distribution of sites with similar properties to the phosphorylated sites (in terms of their presence within the same kinds of protein structural features), but with certainty that they cannot be phosphorylated. Such an analysis of site conservation for pX residues would thus be largely descriptive at present, until we can complete the understanding of how pS/T and pY modifications have evolved, and then understand how pX signalling has evolved.

6. *The numbers of pSer, pThr and pTyr sites identified following UPAX enrichment are quite large, but lower than the numbers typically found using conventional phosphoproteomic approaches on HeLa cell samples. Did the authors split a tryptic digest and carry out a comparison of the numbers of canonical phosphosites obtained by the two methods of enrichment? Do the characteristics of the canonical phosphosites from the UPAX analysis differ in any way from those identified using, for instance, TiO₂ enrichment.*

TiO₂ or IMAC-based phosphopeptide enrichment is extremely effective at depleting non-phosphorylated peptides (and acid-labile phosphopeptides). Minimal instrument time is thus spent analysing non-phosphorylated peptides compared with the same time investment analysing UPAX fractions, which serves to separate, rather than enrich, phosphopeptides (canonical and non-canonical) from non-phosphopeptides. Making a direct comparison of the total number of pS/T/Y sites between the two methods is therefore not pertinent here.

To address the reviewer’s second comment, and evaluate the unbiased nature of the UPAX method presented in this work, the identified UPAX S/T/Y-phosphoproteins were compared to the phosphoproteins identified by an alternative enrichment strategy reported in the literature. The work by Sharma *et al.* (Sharma *et al.* Cell Reports, 2014) describes the phosphoproteins identified from HeLa lysates that were first fractionated by SCX then enriched for phosphopeptides using TiO₂. This two-stage fractionation and enrichment approach employed for deep phosphoproteome coverage results in a greater number of pS/T/Y-containing phosphoprotein IDs (5545 proteins) compared with the 2074 unique pS/T/Y-containing phosphoproteins identified here. For comparison, we selected those proteins in the Sharma *et al.* dataset identified under basal (non-stimulated) conditions containing at least one ‘Class I’ S/T/Y-phosphosite (*i.e.* site localisation score >0.75 identified with Andromeda/A-score). Even though a different search engine was employed, 77% of the pS/T/Y proteins identified in our study were observed in the Sharma *et al.* dataset, strongly suggesting that there is no overt bias in the phosphoproteins undergoing analysis.

Other points:

1. *The authors should note that there are at least three reported pHis phosphatases - PHPT1, LHPP and PGAM5 (see ref. 7). In this regard, they could also cite Hindupur *et al.* (Nature 555:678, 2018), who recently reported that loss of LHPP in liver tumors results in elevated levels of pHis-containing proteins.*

This additional reference has now been added to the text.

2. Page 3 bottom: Why is only pThr referenced here? It would be more interesting to compare all canonical (pSer + pThr + pTyr) phosphorylations versus all non-canonical phosphorylations.

This has now been amended in the abstract and at the end of the introduction to reflect our findings that the number of non-canonical phosphorylation sites is approximately one-third of that observed for canonical phosphorylation, even having accounted for potential false localisation.

3. Page 4 top: Hydrolysis rate and half-life are also strongly dependent on temperature, and so these values will change over the range from 40C to 90oC. For example, at pH 4 at 90oC, 100% of signal was degraded in 10 min. The temperature used is indicated in the Materials and Methods (25oC), but it would be important to consider this aspect within the main text.

These comments appear to be contradictory. We are unsure to which data (complete hydrolysis at pH4 at 90 °C in 10 min) the reviewer refers. We would like to clarify that the experiments evaluating pH-dependence of pHis peptide stability were performed at 25 °C. We had not previously included our analysis of temperature dependence (performed at pH 7.2) but have now added this data to Appendix Fig S1. As we did not perform any experiments with both elevated temperature and reduced pH, it is unclear where and why a combination of these factors should be mentioned in the text.

4. Page 4: What do the authors know about the 1/3-pHis identity of the His sites phosphorylated chemically in myoglobin with phosphoramidate? Is it clear that myoglobin Lys residues were not also phosphorylated under these conditions?

We have now added a dot blot (Fig. EV1) and relevant text to describe the isomeric configuration of pHis following potassium phosphoramidate treatment of myoglobin using published pHis isomer specific antibodies and identify this to be primarily in the 3-pHis form. Under the conditions used here for chemical phosphorylation we did not detect phosphorylation of any residue other than the pHis detailed in Fig EV2.

5. Page 4 bottom: Does this mean that the elution is time dependent or volume dependent? Does a longer period (lower flow rate) with TEA elute phosphopeptides in earlier fractions? Are the peptides identified in early fractions only non-phosphorylated peptides from incomplete chemical phosphorylation or did they lose their phosphate during the whole process by hydrolysis as shown in Fig. EV3? If the phosphopeptides are eluted in a time-dependent manner, then the fact that the later fractions that eluted at an increased pH contained all the phosphopeptides might mean that increased pH reduces the hydrolysis.

As with all IEX, this is dependent on concentration of salt, not time or volume. Data in EV3 demonstrates that some non-phosphorylated His peptides identified in fractions before and after the pHis equivalent, indicating on-column hydrolysis at pH 6.0 that is peptide dependent.

6. Page 5: The authors should discuss the potential issues with the LC-MS/MS loading and resolving buffers being rather acidic, and how this might affect the recovery of phosphopeptides with non-canonical phosphoamino acids, due to the potential for phosphate loss during loading and chromatographic resolution at pH ~2.5.

We cannot rule out that some hydrolysis of acid-labile phosphopeptides will occur over the time taken to perform the LC-MS/MS analysis. However, loading onto the analytical column is performed at ~pH 6.5 and the peptides are only subjected to ~pH 2.8 at the point

that chromatographic elution is initiated. If on-column hydrolysis was significant, then we would identify significant levels of non-phosphorylated pHis Myo peptides in all SAX fractions which we do not (see e.g. Fig.1). Only ~3.4% of the non-canonical peptides identified at *ptmRS* 0.90 were observed in their non-phosphorylated form in the same SAX fraction (suggesting on-column phosphopeptide hydrolysis), indicating that there is negligible effect of the LC-MS conditions on phosphate loss in these experiments.

7. Page 5: It appears that the pH gradient elution using TEA helped to separate unphosphorylated peptides from phosphopeptides in the later fractions. This suggests that nearly no phosphates were lost during the ionization process at pH 2 and higher temperature when injected into the instrument, since 90-100% of peptides phosphorylated were obtained in these fractions. How do the authors explain this surprising stability? Is this a result of short loading/injection times (c.f. Potel et al. ref. 26)?

The time taken for nanoelectrospray ionisation is orders of magnitude lower than the half-life for pHis peptide hydrolysis, therefore we would not expect the elevated temperatures, or acidic pH over this duration to have any noticeable effect on pHis stability (or for the other labile phosphopeptides) and thus detection (please also see previous comment). Indeed, we and others (Gonzalez-Sanchez, *et al.* IJMS 2014; Lin *et al* MCP 2018; Trentini *et al* MCP 2014; Penkhert JASMS 2019) have demonstrated that it is possible to readily detect acid-labile phosphopeptides post-synthesis by LC-MS/MS under ‘standard’ ionisation conditions.

8. Page 7: The strategy is interesting, but it is not really clear whether the final list of proteins is from pSer/Thr/Tyr peptide site identification by MS/MS or phosphopeptide identification by MS. What do the authors mean by "first pass search" here? It will be totally different as the mass of a tryptic peptide + phosphate will not change regardless of the position of the phosphorylated residue, but the theoretical mass of a fragmented ion pS/T/Y (MS/MS) will change the m/z ratio as a function of the phosphoresidue and increase the combinations that can bias the final option even with a high resolution instrument.

Apologies if our database search strategy was not clearly described. The first pass ‘seed’ search was used to generate a smaller database of observable human proteins from this type of cell lysate preparation (asynchronous HeLa cells, Urea-based extraction buffer). The ‘first pass’ search was performed only considering pS/T/Y as variable modifications to minimise informatics search space in the more complex search for all variable phosphosites. Those proteins that were identified from this minimal phosphorylated residue search were then used to generate a smaller database (7,254 entries as opposed to the 20,187 proteins in the UniProt human reviewed database) and used to search for both **non-canonical and canonical phosphorylation**, on the assumption that any phosphate-containing proteins would also yield another non-phosphorylated or pS/T/Y containing peptides that would be identified in this first pass ‘seed’ search. It is feasible that some of the pS/T/Y phosphopeptides identified in the first pass search are incorrectly localised pX peptides, however, only protein level information was used at this stage. All tandem MS spectra were searched using the smaller human proteins database considering all possible permutations of phosphorylation; the site of phosphorylation in the ‘first pass search’ thus did not influence the final phosphosite identification reported. We have added some additional information to the Methods for clarification.

9. Page 7: Is there a final listing for each unique phosphoresidue? For example, do the 225 unique pHis sites come from the 289 His sites in the list of phosphopeptides >0.75 (c.f. EV1) once the low probability ones are excluded that were identified as containing another phosphoresidue at >0.75 on the same peptide?

We have added an additional expanded view dataset (EV2) that details the protein level information of identified phosphorylation sites at each site localisation score. We have also added additional filtering criteria to dataset EV1 to enable filtering by each pX residue.

10. Page 8: Did the authors use a stochastic model with all the potential phosphoresidues from their "limited human database" or did they consider the full human proteome? (See comments on Material and Methods section)

As explained above, to increase our statistical power, we performed a two-pass search, by first discovering the potential sets of proteins supported by the data (searching for pSTY as variable modifications only), we then used that smaller database to perform the search for all possible sites on which we wished to discover evidence for phosphorylation.

11. Page 17: In line 7 and in the second paragraph of the section on "Phosphoarginine", the authors say pLys, when presumably they mean pArg?

Thanks to the reviewer for pointing out this typo – this has now been corrected.

12. Page 18: Do any of the pCys sites correspond to the active site peptides from proteins like PTPs, which are known to have a pCys intermediate?

We have interrogated all of our pX data for sites aligning with known active site phosphate intermediates using UniProt, but did not identify any overlap.

13. Page 21: Was the 28% number derived from the limited human proteome database used? Otherwise these 6 residues should represent more than 60% of the entire human proteome sequence.

Following this query by the reviewer, we have re-calculated the numbers, and indeed we are correct – His, Asp, Glu, Lys, Arg and Cys account for 28% of the amino acid content of the redundant Human Proteome as define in UniProt.

Materials and Methods:

1. Page 7: As specified by the supplier, hydroxyapatite should never be centrifuged, because it explodes the resin beads and strongly decreases efficiency, especially if an acid condition is used.

Having gone through the documentation received with our hydroxyapatite, and checked directly with the supplier, there are no constrains or concerns about centrifuging this resin under the conditions defined herein (1 min, 3000 x g). The pH conditions used (above pH 5) are also within the acceptable limits of this bio-gel.

2. Page 7: Hydroxyapatite acts as an affinity resin, and so a longer incubation as with antibody binding is not necessary.

We thank the review for the comment, however, extended incubation time as performed here will not influence the conclusions from our HAP experiments.

3. Page 7: The idea of using pAla and normalizing to the numbers of each amino acid is innovative, but the rationale or relative frequencies will be different for every peptide according to the different sequence, meaning that in principle every peptide should be normalized on its own sequence. Even if every peptide is considered independently, if one imagines a scenario with the same residue at the C-terminus and N-terminus and thus with at least a twofold higher chance to be phosphorylated than any unique site between, there is still zero risk to position the phosphate at the opposite position with the ion series as it

will be necessarily fragmented. It is for this reason that small peptides (less than 6-7 amino acid) are usually not considered

We respectfully disagree with the reviewer on this point. The pAla FLR estimation method is a global correction method rather than a local one. As we state in the manuscript, it is based on an assumption that all amino acids have an equal probability to be assigned a phosphorylation site *at random*. The reviewer makes the point that within each peptide there are variability probabilities associated with the chance of assigning a phosphosite to a given residue, based on the count of that residue in the peptide. As a hypothetical example, if a phosphopeptide had the sequence EEEEEESK on one phosphosite, there are indeed many more chances to get a pGlu rather than a pSer assignment. However, our method effectively averages across all peptides to make an estimation of global FDR, taking into account the frequency of amino acids in phosphopeptides. Asp and Glu amino acids are observed more frequently in phosphopeptides than other amino acids, and hence they are “penalised” by having higher FLR estimates than pHis. *ptmRS* already performs local weighting, so a peptide with more possible sites tested will have weaker scores (and lower chance of significance) than a peptide with few sites considered, and thus does not need to be down-weighted further in the global re-scoring.

4. Page 8: The chance of mislocalizing the position of phosphate after MS/MS is more likely due to the space between two potential residues. So if the frequencies of residues are considered for mathematical correlation, a risk factor of distance between these residue should be also integrated.

As noted above, this is implicitly considered in *ptmRS* scoring, since there are fewer potential ions that can differentiate sites that are closer within a peptide than more distant. We are confident that the method is robust and do not believe there is a straightforward method to further correct for this possibility.

Yet, another aspect is that His is the only residue that can generate two phosphoforms (1- and 3-pHis) - does this mean that the FLR should be divided by two when this will likely depend on the steric hindrance and the tertiary structure of the protein.

There is currently no evidence that the HCD fragmentation pattern, or likelihood of site localisation differs for the two pHis isoforms, and indeed we demonstrate that we are able to identify pHis sites on proteins previously been reported to contain either 1-pHis or 3-pHis and that our analysis does not discriminate. Consequently, we are not considering these as separate events and FLR is only computed on the basis of (p)His frequency.

5. Page 8: Is the C-terminal phosphoresidue normalized by the non-C-terminal calculation or are they not normalized at all because the Ala can never be at the C-terminal position?

There are no peptide C-terminal sites reported in the final datasets, since we believe it unlikely that trypsin can cleave pLys or pArg residues. The pLys and pArg FLR is estimated via the frequency of internal Lys and Arg residues only.

6. Page 9: How do the authors explain that, considering the normalization, there is an increase in pHis signal for pH 4/pH 6 at $t = 15$ min and $t = 1$ h compared to $t = 0$? If this is from free phosphoramidate, then the chemical phosphorylation would still be ongoing, meaning that one is not defining the degradation, but rather turnover.

Free phosphoramidate has been removed by C18 clean-up (under non-acidic conditions) of the pHis Myo peptides following digestion. The slight increase (17%) in pHis peptide levels is within the quantitative variability of this particular assay.

7. Page 10: Does the 33% increase in non-phosphorylated peptides after neutral loss

correspond to 100% of phosphorylated peptides or 1%? What was the original ratio of phosphorylated versus non-phosphorylated peptides in the load? Were there some pHis-containing peptides that were eluted from the resin, but potentially lost their phosphate subsequently by hydrolysis?

Due to the difference in ionisation efficiency between phosphorylated and non-phosphorylated peptides, it is not possible to determine exactly what the stoichiometry of phosphorylation is based on relative signal. However, the relative signal intensity might suggest in the order of between 30-60% phosphorylation for the individual phosphosites following phosphoramidate treatment. Although it is possible that the pHis peptides were eluted from SAX and subsequently hydrolysed, we believe this to be extremely unlikely based two observations: i) pHis peptides are readily observed in the absence of TiO₂ enrichment, and ii) identification of the non-phosphorylated equivalents of phosphopeptides containing pHis in the same SAX fraction following UPAX was <3%.

8. Page 18: Were there any eluted His-containing peptides lacking phosphate identified that corresponded to identified pHis peptides?

Please see previous responses to this and the previous reviewer.

Minor points: 1. Abstract: Cys phosphorylation should be mentioned in the Abstract.

This has been amended.

2. Keywords: Why choose to specify phosphoaspartate as a key word and not any of the other non-canonical phosphoaminoacids?

We would happily mention all types of non-canonical phosphorylation, but unfortunately only five keywords are permitted. We have amended the keywords to include only phosphohistidine as we think that this will be of most immediate relevance.

Referee #3:

Comments

this manuscript is a valuable addition to the phospho-proteomics field by introducing a novel approach to discover unconventional phosphorylation sites. A few key questions are well addressed, such as false discovery rate and motifs. That's being said, I have a few additional questions and comments to the authors.

Major comments:

1. A main experiment missing from this manuscript is negative control. Although the authors have an attempt to knock down histidine phosphatase, no apparent difference in the number of pHis is observed. An alternative control could be acid-treated or heat-treated peptide digests. Isotopic/isobaric labeling may be applied to the acid-treated and untreated peptides, combined and separated on UPAX. A true non-canonical phosphosite should demonstrate decreased quantitative signal due to hydrolysis. This would provide a strong support to the identity of reported sites.

Unfortunately the experiment suggested by the reviewer is challenging and likely to be subject to misinterpretation. Although we have presented here data on the acid (and now heat) stability of a set of pHis-containing peptides, the inability to generate peptide standards for many of the other non-canonical phosphorylation sites means that it is not possible to perform the analogous studies for all the other non-canonical phosphorylation sites. Consequently, data describing the pH and heat dependency of the individual pX

residues is not currently available and the data would be subject to (potentially significant) misinterpretation, as there is no guarantee that the treatment would indeed eliminate all non-canonical pX sites. Secondly, hydrolysed phosphopeptides would chromatograph differently by SAX. This would change the complexity of the latter phosphopeptide-rich SAX fractions quite considerably (assuming that hydrolysis was extensive), potentially resulting in differential ionisation, which could compromise the ability to directly compare the results of the 'before' and 'after' treatment.

We agree with the reviewer that an additional negative control would be beneficial and therefore present results from a global TiO₂-based phosphopeptide enrichment experiment, where sample preparation and phosphopeptide enrichment was performed under standard conditions, but LC-MS/MS data acquisition and database searching was performed as described for UPAX. This data is now presented in Fig. 3C (and Appendix Table S7) and discussed in the main text. As the reviewer will note, at *ptmRS* 0.90, the proportion of non-canonical phosphosites compared with pSer/pThr/pTyr decreases from 35% using the UPAX strategy to 2.9% with a standard phosphopeptide preparation strategy, demonstrating that the increased prevalence of non-canonical phosphorylation sites we see with UPAX is not due to a difference in data acquisition or data interrogation strategies.

2. What is the identification rate of unconventional phosphosites on a TiO₂ enriched global phospho data set when modification is permitted on H/K/R/D/E/C as well as S/T/Y during mascot database search? How is it compare to UPAX approach?

Please see comments above.

3. It's not entirely clear to me that how three biological replicates are summarized, i.e. only the sites identified in all three are retained in the final set, or observation in any replicates are included in the final set?

Phosphopeptides identified in either of the two conditions (NT siRNA or PHPT1 siRNA) across any of the three biological replicates per condition are retained. The number of instances of observation is clearly detailed in datasets EV1 and EV2.

4. High frequency of c-terminal lysine phosphorylation is a concern to me. Is it the true false positive rate? Such frequency greatly exceeds pAla estimated FLR rate. I'd like to see more analysis or validation on this subset, rather than just excluding from further analysis. For example, can these sites still be identified when an enzyme with different specificity is used?

We are in the process of generating analogous datasets using alternative enzymes, including GluC and LysN. Unfortunately, this study is still ongoing and will require additional time to complete. Until we can better understand why phosphorylation sites are being localised to these C-terminal residues, we think that the best option is to remove these sites, as has been done with other analogous datasets e.g. Trentini MCP 2014.

Minor comments:

1. Figure 1 A-C: how many peptides are summarized here? Please indicate it in the figure legend.

This information has now been included in the figure legend.

2. Figure 2 A indicates HeLa cells 24h but figure legend shows 48h. Please confirm.

The information in the figure legend has now been corrected to 24 h.

3. Page 17: ~107 pLys sites Should be pArg sites.

This has now been corrected.

2nd Editorial Decision

19th July 2019

Thank you for submitting your revised version on UPAX-identification of non-canonical phosphorylation for our consideration. Referees 1 and 2 have now assessed it once more, and I am pleased to say that both of them consider the paper improved and now in principle suitable for publication. However, they still retain a few specific concerns, of which at least those raised by referee 2 as well as the first point of referee 1 would still require addressing prior to acceptance. I am therefore returning the paper to you for an additional, final round of minor revision.

REFeree REPORTS

Referee #1:

I am now happy with the paper. I thank the authors for responding to all of my questions.

I have two comments concerning requests of the other two reviewers:

1. phosphosites are usually not well evolutionarily conserved as the majority of them are in loops which have little evolutionary pressure. Thus, it is unlikely that they will be conserved.
2. an experiment providing a good answer to the first question by reviewer 3 could be to take one UPAX fraction with highly enriched phosphopeptides and split it in half. one half is then heat or acid-treated and both are analysed by their LC-MS approach. any acid-labile phosphosites should be reduced/gone

Referee #2:

In the revised version, the authors have added a modest amount of new data. made some changes to filters from their Excel files and improved the figure formatting, which strengthen the paper.

1. There is a problem with the new anti-pHis dot blot data in Figure EV1. Phospho-NME1 should be recognized by the anti-1-pHis antibodies, whereas the phospho-PGAM should be recognized by the anti-3-pHis antibodies. However, as shown, they detected pNME1 with the 3-pHis mAb and PGAM with the 1-pHis mAb. Possibly, the labels of the samples in lanes 1 and 2 were swapped, but this needs to be addressed.
2. Despite the authors' claims to have eliminated identified peptides with non-canonical pX sites containing C-terminal pK and pR sites, there still appear to be a number peptides in the dataset with pK or pR at the C-terminus, e.g. #9 Q96QE2 in database 1505965.

Referee #1:

1. phosphosites are usually not well evolutionarily conserved as the majority of them are in loops which have little evolutionary pressure. Thus, it is unlikely that they will be conserved.

It is unclear what the basis is of the referee's statement that phosphosites are not evolutionarily conserved. Many mechanistically-important phosphorylation sites are very highly conserved on loops (e.g. kinase regulatory sites in the activation segment from yeast to man). Pedro Beltran has shown that there are numerous conserved phosphorylation sites when large numbers of protein domains are considered

(<http://europepmc.org/abstract/MED/31036831>). We (Prof. Andy Jones) have also recently published a paper describing high conservation of phosphomotifs (and functions) between Arabidopsis and rice (over 100 Mya)

(<https://www.ncbi.nlm.nih.gov/pmc/articles/PMC5971217/>).

Moreover, our ongoing investigations (manuscript in preparation) would suggest that many Ser/Thr/Tyr residues that are modified by the addition of phosphate may actually be more conserved across the animal kingdom than Ser/Thr/Tyr residues in the same protein in other domains for which there is no current evidence of phosphorylation e.g. in DNA binding proteins.

As there is no specific mention of non-canonical phosphosite conservation in the manuscript, no additional text has been added to address this comment.

Referee #2:

1. There is a problem with the new anti-pHis dot blot data in Figure EV1. Phospho-NME1 should be recognized by the anti-1-pHis antibodies, whereas the phospho-PGAM should be recognized by the anti-3-pHis antibodies. However, as shown, they detected pNME1 with the 3-pHis mAb and PGAM with the 1-pHis mAb. Possibly, the labels of the samples in lanes 1 and 2 were swapped, but this needs to be addressed.

We thank the referee for pointing out this labelling error; this has now been rectified.

2. Despite the authors' claims to have eliminated identified peptides with non-canonical pX sites containing C-terminal pK and pR sites, there still appear to be a number peptides in the dataset with pK or pR at the C-terminus, e.g. #9 Q96QE2 in database 1505965.

Apologies, an old version of the EV datasets were inadvertently uploaded at revision stage. The correct EV datasets are now included with this revised version of the submission; C-terminal K/R phosphosites have been filtered (although not removed for transparency) from the datasets.

Thank you for submitting your final revised manuscript for our consideration. I am pleased to inform you that we have now accepted it for publication in The EMBO Journal.

USEFUL LINKS FOR COMPLETING THIS FORM

Corresponding Author Name: Claire E. Eyers
Journal Submitted to: EMBO Journal
Manuscript Number: EMBOJ-2018-100847R